# Understanding and Improving Shampoo and SOAP via Kullback–Leibler Minimization

Wu Lin[1,*], Scott C. Lowe[1], Felix Dangel[1], Runa Eschenhagen[2], Zikun Xu[3], and Roger B. Grosse[1,4]

[1]Vector Institute, Toronto, Canada
[2]University of Cambridge, Cambridge, United Kingdom
[3]Microsoft, United States
[4]University of Toronto, Toronto, Canada

## Abstract

Shampoo and its efficient variant, SOAP, employ structured second-moment estimations and have shown strong performance for training neural networks (NNs). In practice, however, Shampoo typically requires step-size grafting with Adam to be competitive, and SOAP mitigates this by applying Adam in Shampoo's eigenbasis—at the cost of additional memory overhead from Adam in both methods. Prior analyses have largely relied on the Frobenius norm to motivate these estimation schemes. We instead recast their estimation procedures as covariance estimation under Kullback-Leibler (KL) divergence minimization, revealing a previously overlooked theoretical limitation and motivating principled redesigns. Building on this perspective, we develop **KL-Shampoo** and **KL-SOAP**, practical schemes that match or exceed the performance of Shampoo and SOAP in NN pre-training while achieving SOAP-level per-iteration runtime. Notably, KL-Shampoo does not rely on Adam to attain competitive performance, eliminating the memory overhead introduced by Adam. Across our experiments, KL-Shampoo consistently outperforms SOAP, Shampoo, and even KL-SOAP, establishing the KL-based approach as a promising foundation for designing structured methods in NN optimization.

## 1 Introduction

Optimizer Shampoo (Gupta et al., 2018) has recently rivaled and, in several benchmarks, surpassed Adam (Kingma & Ba, 2015) in training a wide range of neural network (NN) models (Dahl et al., 2023; Kasimbeg et al., 2025). Consequently, Shampoo and its efficient variant, SOAP (Vyas et al., 2025a), have drawn increasing attention. In practice, however, Shampoo typically requires step-size grafting with Adam to achieve competitive performance (Agarwal et al., 2020; Shi et al., 2023). SOAP mitigates this by applying Adam in Shampoo's eigenbasis and further reducing per-iteration runtime. Unfortunately, this reliance on Adam introduces additional memory overhead in both methods. Prior work (Morwani et al., 2025; Eschenhagen et al., 2025; An et al., 2025; Xie et al., 2025) has investigated their structural preconditioner schemes—which approximate the flattened gradient second moment (Duchi et al., 2011)—primarily through the Frobenius norm. However, few studies have examined these schemes from the perspective of Kullback–Leibler (KL) divergence. Compared to the Frobenius norm, the KL divergence between zero-mean Gaussian covariance matrices is more appropriate for interpreting Shampoo's and SOAP's preconditioners as Gaussian covariance estimators, since the second moment they approximate can be viewed as the covariance matrix of a zero-mean Gaussian. Historically, the Frobenius norm (Greenstadt, 1968; Dennis & Moré, 1977) has been replaced by the KL divergence (Fletcher, 1991) or its second-order truncation (Nocedal & Wright, 2006) as the foundation for designing preconditioner estimation schemes in quasi-Newton methods. The KL perspective has provided a unified framework for interpreting (Güler et al., 2009; Waldrip & Niven, 2016) and extending (Kanamori & Ohara, 2013a;b) structural preconditioner estimation in quasi-Newton methods such as BFGS and DFP—something the Frobenius norm does not. Moreover, the KL divergence intrinsically respects the symmetric positive-definite (SPD) constraint (Amari, 2016;

---

*Corresponding author: yorker.lin@gmail.com; Code: github.com/yorkerlin/KL-Methods

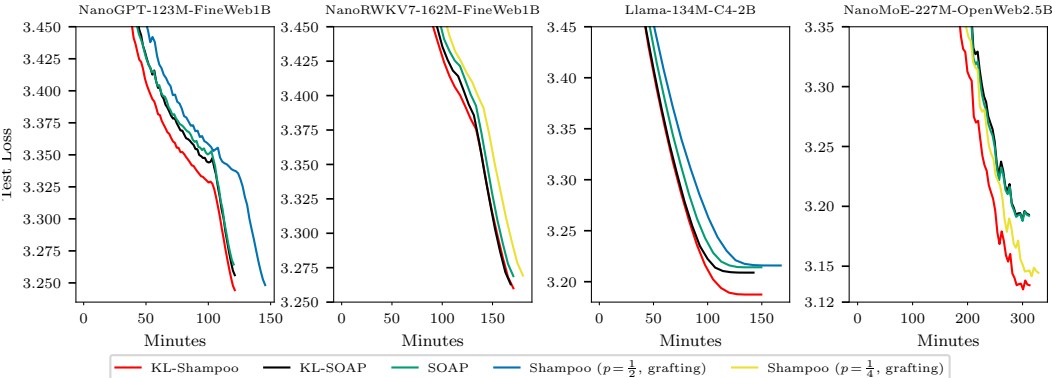

Figure 1: Empirical results (random search with 150 runs per method) on language models using bfloat16 demonstrate the advantages of KL-based methods over Shampoo and SOAP, while matching SOAP's per-iteration runtime. All methods take the same number of iterations in these experiments. Surprisingly, KL-Shampoo outperforms KL-SOAP. We include the best Shampoo run based on a state-of-the-art implementation from Meta (Shi et al., 2023). See Fig. 9 in Appx. H for evaluating KL-Shampoo on a larger model (Llama3 with 450M).

Minh & Murino, 2017) that preconditioners in Shampoo and SOAP must satisfy as preconditioned methods (Nesterov et al., 2018)—a property the Frobenius norm lacks. This constraint implies that the entries of the preconditioning matrix do not play equivalent roles and therefore should not be treated equally (Pennec et al., 2006; Bhatia, 2007)—a point the Frobenius norm ignores.

In this work, we introduce a KL perspective that interprets the estimation schemes of Shampoo and SOAP as solutions to KL-minimization problems for covariance estimation, thereby connecting structured adaptive methods with both classical quasi-Newton ideas and structural Gaussian whitening within a single framework. Our framework naturally extends to tensor-valued settings, where some existing theoretical interpretations based on singular value decomposition (SVD) or the spectral norm may not apply. This perspective reveals a key limitation (illustrated in Fig. 5): the Kronecker-structured estimators used by Shampoo and SOAP do not adequately solve the corresponding KL-minimization problem. This limitation, in turn, opens new opportunities for improvement. Leveraging this insight, we refine the estimation rules of Shampoo and SOAP and develop practical KL-based schemes—**KL-Shampoo** and **KL-SOAP**—that match or exceed the performance of Shampoo and SOAP for NN (pre-)training while maintaining SOAP-level per-iteration runtime. Notably, KL-Shampoo does not rely on Adam to achieve competitive performance, thereby avoiding Adam's additional memory overhead (Table 1). We then generalize our framework as a divergence-based estimation approach and consider other Shampoo variants, including a trace-scaling variant—a matrix version of Adafactor. In addition, the practical techniques we develop for KL-Shampoo (Sec. 4) can be adapted to strengthen Shampoo variants and make the trace-scaling variant competitive without step-size grafting (Fig. 10, Appx. H), while achieving SOAP-level per-iteration runtime (Fig. 11, Appx. H). Empirically, we show that KL-based methods are competitive for training a range of NNs and remain as flexible as Shampoo and SOAP for tensor-valued weights. Surprisingly, KL-Shampoo consistently outperforms other Shampoo variants in our experiments (Figs. 1, 5 and 9). Overall, our approach provides a principled and unifying way to design structured preconditioned methods for NN optimization (see Sec. 3.3).

## 2 BACKGROUND

**Notation** For presentation simplicity, we focus on matrix-valued weights and the optimization update for a single parameter matrix $\boldsymbol{\Theta} \in \mathcal{R}^{d_a \times d_b}$, rather than a set of weight matrices for NN training. We use $\mathrm{Mat}(\cdot)$ to unflatten its input vector into a matrix and $\mathrm{vec}(\cdot)$ to flatten its input matrix into a vector. For example, $\boldsymbol{\theta} := \mathrm{vec}(\boldsymbol{\Theta})$ is the flattened weight vector and $\boldsymbol{\Theta} \equiv \mathrm{Mat}(\boldsymbol{\theta})$ is the unflattened weight matrix. Vector $\boldsymbol{g}$ is a (flattened) gradient vector for the weight matrix. We denote $\gamma$, $\beta_2$ and $\boldsymbol{S}$ to be a step size, a weight for moving average, and a preconditioning matrix for an adaptive method, respectively. $\mathrm{Diag}(\cdot)$ returns a diagonal matrix whose diagonal entries are given by its input vector, whilst $\mathrm{diag}(\cdot)$ extracts the diagonal entries of its input matrix as a vector.

**Shampoo** Given a matrix gradient $\boldsymbol{G}$ and the flattened gradient $\boldsymbol{g} = \mathrm{vec}(\boldsymbol{G})$, the original Shampoo method (Gupta et al., 2018) considers a *Kronecker-factored* approximation, $(\boldsymbol{S}_a)^{2p} \otimes (\boldsymbol{S}_b)^{2p}$, of the flattened gradient second moment, $\mathbb{E}_{\boldsymbol{g}}[\boldsymbol{g}\boldsymbol{g}^\top]$, where $p$ denotes a matrix power, $\boldsymbol{S}_a := \mathbb{E}_{\boldsymbol{g}}[\boldsymbol{G}\boldsymbol{G}^\top]$, $\boldsymbol{S}_b := \mathbb{E}_{\boldsymbol{g}}[\boldsymbol{G}^\top\boldsymbol{G}]$, and $\otimes$ denotes a Kronecker product. In practice, we often approximate the expectation, $\mathbb{E}_{\boldsymbol{g}}[\boldsymbol{g}\boldsymbol{g}^\top]$, with an exponentially moving average (EMA) on the outer product (Morwani et al., 2025). The original Shampoo method uses the $1/4$ power (i.e., $p = 1/4$) and other works (Anil et al., 2020; Shi et al., 2023; Morwani et al., 2025) suggest using the $1/2$ power (i.e., $p = 1/2$). At each iteration, Shampoo follows this update rule with EMA on $\boldsymbol{S}_a$ and $\boldsymbol{S}_b$:

$$\boldsymbol{S}_a \leftarrow (1 - \beta_2)\boldsymbol{S}_a + \beta_2\boldsymbol{G}\boldsymbol{G}^\top, \quad \boldsymbol{S}_b \leftarrow (1 - \beta_2)\boldsymbol{S}_b + \beta_2\boldsymbol{G}^\top\boldsymbol{G} \quad \text{(Kronecker 2}^{\text{nd}}\text{ moment est.)},$$

$$\boldsymbol{\theta} \leftarrow \boldsymbol{\theta} - \gamma\boldsymbol{S}^{-1/2}\boldsymbol{g} \iff \boldsymbol{\Theta} \leftarrow \boldsymbol{\Theta} - \gamma\boldsymbol{S}_a^{-p}\boldsymbol{G}\boldsymbol{S}_b^{-p} \quad \text{(Preconditioning)}, \tag{1}$$

where $\boldsymbol{S} := \boldsymbol{S}_a^{2p} \otimes \boldsymbol{S}_b^{2p}$ is Shampoo's preconditioning matrix, and we leverage the Kronecker structure of $\boldsymbol{S}$ to move from the left expression to the right expression in the second line.

> **Shampoo's implementation employs eigendecomposition.** Shampoo is typically implemented by using the eigendecomposition of $\boldsymbol{S}_k$, such as $\boldsymbol{Q}_k\mathrm{Diag}(\boldsymbol{\lambda}_k)\boldsymbol{Q}_k^\top = \mathrm{eigen}(\boldsymbol{S}_k)$, for $k \in \{a, b\}$, every few steps and storing $\boldsymbol{Q}_k$ and $\boldsymbol{\lambda}_k$ (Anil et al., 2020; Shi et al., 2023). Therefore, the power of $\boldsymbol{S}_k$ is computed using an elementwise power in $\boldsymbol{\lambda}_k$ such as $\boldsymbol{S}_k^{-p} = \boldsymbol{Q}_k\mathrm{Diag}(\boldsymbol{\lambda}_k^{\odot -p})\boldsymbol{Q}_k^\top$, where $\cdot^{\odot p}$ denotes elementwise $p$-th power. This computation becomes an approximation if the decomposition is not performed at every step.

> **Using Adam for Shampoo's stabilization increases memory usage.** If the eigendecomposition is applied infrequently to reduce iteration cost, Shampoo has to apply step-size grafting with Adam to maintain performance (Agarwal et al., 2020; Shi et al., 2023) as empirically shown in Fig. 2. Unfortunately, this increases its memory usage introduced by Adam (see Table 1).

**SOAP** SOAP improves Shampoo with the $p = 1/2$ power by running Adam in the eigenbasis of Shampoo's preconditioner $(\boldsymbol{S}_a)^{2p} \otimes (\boldsymbol{S}_b)^{2p} = \boldsymbol{S}_a \otimes \boldsymbol{S}_b$. Notably, SOAP reuses Shampoo's Kronecker estimation rule for computing the eigenbasis $\boldsymbol{Q} := \boldsymbol{Q}_a \otimes \boldsymbol{Q}_b$ and incorporates Adam's $2^{\text{nd}}$ moment, denoted by $\boldsymbol{d}$, for preconditioning, where $\boldsymbol{Q}_k$ is Shampoo's Kronecker eigenbasis $\boldsymbol{S}_k$ for $k \in \{a, b\}$ defined above. As a result, SOAP effectively employs an *augmented* preconditioner, $\boldsymbol{S} := \boldsymbol{Q}\mathrm{Diag}(\boldsymbol{d})\boldsymbol{Q}^\top$, which cannot be expressed as a Kronecker product of any two matrices with the same shape as $\boldsymbol{S}_a$ and $\boldsymbol{S}_b$. Because we omit momentum (i.e. let Adam's $\beta_1 = 0$), SOAP takes the following step with the Adam update becoming an RMSProp update (Tieleman & Hinton, 2012):

$$\boldsymbol{S}_a \leftarrow (1 - \beta_2)\boldsymbol{S}_a + \beta_2\boldsymbol{G}\boldsymbol{G}^\top, \quad \boldsymbol{S}_b \leftarrow (1 - \beta_2)\boldsymbol{S}_b + \beta_2\boldsymbol{G}^\top\boldsymbol{G} \quad \text{(Shampoo's 2}^{\text{nd}}\text{ moment est.)},$$

$$\boldsymbol{d} \leftarrow (1 - \beta_2)\boldsymbol{d} + \beta_2\hat{\boldsymbol{g}}^{\odot 2} \quad \text{(RMSProp's diagonal 2}^{\text{nd}}\text{ moment est. in the eigenbasis)},$$

$$\boldsymbol{\theta} \leftarrow \boldsymbol{\theta} - \gamma\boldsymbol{S}^{-\frac{1}{2}}\boldsymbol{g} \iff \boldsymbol{\Theta} \leftarrow \boldsymbol{\Theta} - \gamma\boldsymbol{Q}_a^\top \mathrm{Mat}\left(\frac{\hat{\boldsymbol{g}}}{\sqrt{\boldsymbol{d}}}\right)\boldsymbol{Q}_b \quad \text{(Preconditioning)}, \tag{2}$$

where $\hat{\boldsymbol{g}} := \boldsymbol{Q}^\top\boldsymbol{g} = \mathrm{vec}(\boldsymbol{Q}_a^\top\boldsymbol{G}\boldsymbol{Q}_b)$ is a "projected" gradient vector in eigenbasis $\boldsymbol{Q}$ and recall that $\boldsymbol{S} := \boldsymbol{Q}\mathrm{Diag}(\boldsymbol{d})\boldsymbol{Q}^\top$ is SOAP's preconditioner. Here, we leverage the Kronecker structure and orthogonality of the eigenbasis to move from the left to the right in the last line of Eq. (2). Note that this EMA weight $\beta_2$ is defined as $1 - \beta_2^{(\text{Adam})}$, where $\beta_2^{(\text{Adam})}$ is Adam's (RMSProp's) $\beta_2$. We use this definition rather than Adam's because we want to further interpret this moving-average scheme through the lens of our KL perspective.

> **SOAP's implementation utilizes QR decomposition.** SOAP requires only the eigenbasis, which is approximated via a QR decomposition, whereas Shampoo typically employs an eigendecomposition to compute both the eigenbasis and the eigenvalues. Vyas et al. (2025a) therefore suggest replacing the slower eigendecomposition with the faster QR decomposition, such as $\boldsymbol{Q}_k \leftarrow \mathrm{qr}(\boldsymbol{S}_k\boldsymbol{Q}_k)$ for $k \in \{a, b\}$. This makes SOAP more computationally efficient than Shampoo.

> **Running Adam in the eigenbasis increases memory usage.** Introducing Adam's (RMSProp's) $2^{\text{nd}}$ moment estimation increases SOAP's memory consumption (see Table 1). This is because this estimation, $\boldsymbol{d} \in \mathcal{R}^{d_a d_b \times 1}$, uses extra memory and cannot be exactly expressed as a Kronecker product of any two vectors with compatible dimensions, such as $\boldsymbol{d} \neq \boldsymbol{r}_a \otimes \boldsymbol{r}_b$, where $\boldsymbol{r}_a \in \mathcal{R}^{d_a \times 1}$ and $\boldsymbol{r}_b \in \mathcal{R}^{d_b \times 1}$.

The original Shampoo's Kronecker estimation rule ($p = 1/4$) (Gupta et al., 2018; Duvvuri et al., 2024) is proposed based on a matrix Loewner bound (Löwner, 1934), while recent estimation rules ($p = 1/2$) (Morwani et al., 2025; Eschenhagen et al., 2025) focus on bounds induced by the Frobenius norm. SOAP reuses Shampoo's Kronecker estimation rule and additionally introduces Adam's 2nd-moment estimation in the eigenbasis (Vyas et al., 2025a). None of these works interpret or motivate their estimations as covariance estimation, thereby missing the opportunity to introduce the KL perspective.

## 3 SECOND MOMENT ESTIMATION VIA KULLBACK–LEIBLER MINIMIZATION

We focus on Shampoo with $p = 1/2$ and show that its second-moment estimation can be viewed as a structured covariance estimation problem solved via Kullback–Leibler (KL) minimization. This perspective reflects the natural connection between the flattened gradient second moment (Duchi et al., 2011) that Shampoo approximates and a covariance matrix. From the KL perspective, we reveal a previously unrecognized limitation of Shampoo's estimation rule: the Kronecker-structured estimators used by Shampoo and SOAP do not adequately solve the corresponding KL-minimization problem. This limitation, in turn, opens new opportunities for improvement. Building on this, we propose a KL-based scheme for Shampoo, which we term the idealized **KL-Shampoo**.

**KL Minimization** For simplicity, we begin by introducing a KL perspective in a matrix-valued case and drop subscripts when referring to the flattened gradient 2nd moment, like $\mathbb{E}[\boldsymbol{g}\boldsymbol{g}^\top] := \mathbb{E}_{\boldsymbol{g}}[\boldsymbol{g}\boldsymbol{g}^\top]$, where $\boldsymbol{g} = \text{vec}(\boldsymbol{G})$ is a flattened gradient vector of a matrix-valued gradient $\boldsymbol{G} \in \mathcal{R}^{d_a \times d_b}$. The goal is to estimate a Kronecker-structured preconditioning matrix, $\boldsymbol{S} = \boldsymbol{S}_a \otimes \boldsymbol{S}_b$, that closely approximates the 2nd moment, where $\boldsymbol{S}_a \in \mathcal{R}^{d_a \times d_a}$ and $\boldsymbol{S}_b \in \mathcal{R}^{d_b \times d_b}$ are both symmetric positive-definite (SPD). Motivated by the natural connection between the second moment and a covariance matrix, we treat these as covariance matrices of zero-mean Gaussian distributions and achieve this goal by minimizing the KL divergence between the two distributions,

---

**KL Perspective for Covariance Estimation**

$$\text{KL}(\mathbb{E}[\boldsymbol{g}\boldsymbol{g}^\top], \boldsymbol{S}) := D_{\text{KL}}(\mathcal{N}(\boldsymbol{0}, \mathbb{E}[\boldsymbol{g}\boldsymbol{g}^\top] + \kappa\boldsymbol{I}) \,\|\, \mathcal{N}(\boldsymbol{0}, \boldsymbol{S}))$$

$$= \frac{1}{2}\big(\log\det(\boldsymbol{S}) + \text{Tr}((\mathbb{E}[\boldsymbol{g}\boldsymbol{g}^\top] + \kappa\boldsymbol{I})\boldsymbol{S}^{-1})\big) + \text{const}, \tag{3}$$

---

where $\mathbb{E}[\boldsymbol{g}\boldsymbol{g}^\top]$ and $\boldsymbol{S}$ are considered as Gaussian's covariance, $\det(\cdot)$ denotes the matrix determinant of its input, and $\kappa \geq 0$ is a damping to ensure the positive-definiteness of $\mathbb{E}[\boldsymbol{g}\boldsymbol{g}^\top] + \kappa\boldsymbol{I}$ if necessary.

**Natural extension to tensor-valued weights** Our framework naturally extends to tensor cases, $\boldsymbol{G} \in \mathcal{R}^{d_a \times d_b \times d_c}$, by considering a preconditioner $\boldsymbol{S} = \boldsymbol{S}_a \otimes \boldsymbol{S}_b \otimes \boldsymbol{S}_c$ to approximate the flattened second moment, $\mathbb{E}[\boldsymbol{g}\boldsymbol{g}^\top]$, where matrix $\boldsymbol{S}_k \in \mathcal{R}^{d_k \times d_k}$ is SPD for $k \in \{a, b, c\}$.

**KL minimization as a divergence-based projection over SPD matrices.** The KL divergence between zero-mean Gaussians coincides (up to a constant factor) with the log-determinant divergence widely used in matrix optimization over *SPD matrices* (Dhillon & Tropp, 2008; Kulis et al., 2009; Sra, 2016), which does not require a zero-mean assumption when viewed purely as a matrix divergence. Thus, we can interpret our KL-minimization problems as projecting a target SPD matrix onto the Kronecker-structured SPD manifold (see Table 2). . This viewpoint naturally extends to settings where the target matrix is not a second moment, such as the curvature matrices used in quasi-Newton methods (Fletcher, 1991; Waldrip & Niven, 2016), thereby reconnecting structured adaptive methods with quasi-Newton ideas.

### 3.1 INTERPRETING SHAMPOO'S ESTIMATION AS COVARIANCE ESTIMATION

Similar to existing works (Morwani et al., 2025; Eschenhagen et al., 2025; Vyas et al., 2025a), we first disable the moving average (i.e., let $\beta_2 = 1$) for our descriptions and focus on Shampoo with power $p = 1/2$, presenting a KL minimization perspective and interpreting its estimation rule from this perspective. We will obtain Shampoo's estimation rule by solving a KL minimization problem.

**Claim 1.** *(Shampoo's Kronecker-based covariance estimation) The optimal solution of KL minimization* $\min_{\boldsymbol{S}_a} \text{KL}(\mathbb{E}[\boldsymbol{g}\boldsymbol{g}^\top], \boldsymbol{S})$ *with a one-sided preconditioner* $\boldsymbol{S} = (1/d_b \boldsymbol{S}_a) \otimes \boldsymbol{I}_b$ *is* $\boldsymbol{S}_a^* = \mathbb{E}[\boldsymbol{G}\boldsymbol{G}^\top]$, *where* $d_k$ *is the dimension of matrix* $\boldsymbol{S}_k \in \mathbb{R}^{d_k \times d_k}$ *for* $k \in \{a, b\}$ *and* $\boldsymbol{G} = \text{Mat}(\boldsymbol{g})$.

*Likewise, we can obtain the estimation rule for* $\boldsymbol{S}_b$ *by considering* $\boldsymbol{S} = \boldsymbol{I}_a \otimes (1/d_a \boldsymbol{S}_b)$.

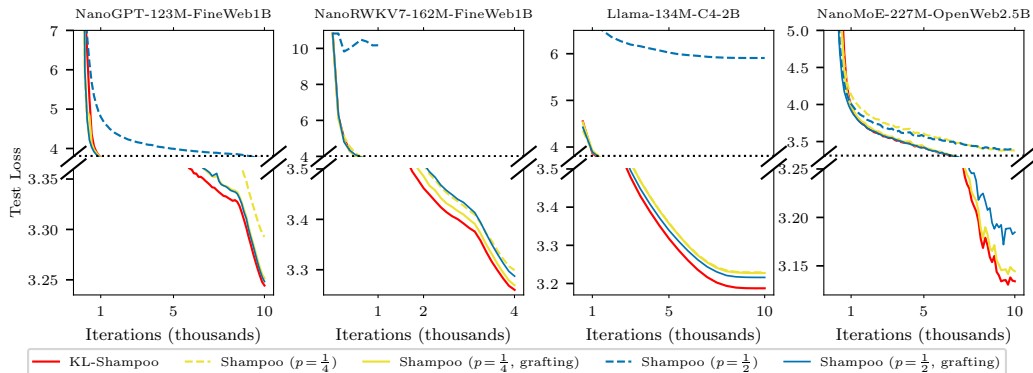

Figure 2: Empirical results (random search with 150 runs per method) on language models using bfloat16 show that KL-Shampoo (with QR decomposition) does not rely on step-size grafting with Adam to perform well. Shampoo without grafting does not perform well, even when using the state-of-the-art implementation (Shi et al., 2023). In particular, Shampoo with power $p = 1/2$ fails to train the RWKV7 model in all 150 runs when grafting is disabled.

|  | Shampoo | SOAP | KL-Shampoo | KL-SOAP |
|---|---|---|---|---|
| Kronecker factors ($\boldsymbol{S}_a$, $\boldsymbol{S}_b$) | $d_a^2 + d_b^2$ | $d_a^2 + d_b^2$ | $d_a^2 + d_b^2$ | $d_a^2 + d_b^2$ |
| Kronecker factors' eigenbasis ($\boldsymbol{Q}_a$, $\boldsymbol{Q}_b$) | $d_a^2 + d_b^2$ | $d_a^2 + d_b^2$ | $d_a^2 + d_b^2$ | $d_a^2 + d_b^2$ |
| Kronecker factors' eigenvalues ($\boldsymbol{\lambda}_a$, $\boldsymbol{\lambda}_b$) | $d_a + d_b$ | N/A | $d_a + d_b$ | $d_a + d_b$ |
| Adam's 2nd moment in the eigenbasis ($\boldsymbol{d}$) 
 (interpreted as augmented eigenvalues, Sec. 5) | N/A | $d_a d_b$ | N/A | $d_a d_b$ |
| Momentum | $d_a d_b$ | $d_a d_b$ | $d_a d_b$ | $d_a d_b$ |
| Step-size grafting with Adam | $d_a d_b$ | N/A | N/A | N/A |

Table 1: Memory usage of each method considered in this work. We store and update [1] all of them in half precision (bfloat16). The memory overhead introduced by Adam is highlighted in red. Note that SOAP's and KL-SOAP's preconditioners, $\boldsymbol{Q}\mathrm{Diag}(\boldsymbol{d})\boldsymbol{Q}^\top$, can not be expressed as a Kronecker product due to the augmented eigenvalues $\boldsymbol{d}$, while Shampoo's and KL-Shampoo's preconditioners, $\boldsymbol{Q}\mathrm{Diag}(\boldsymbol{\lambda}_a \otimes \boldsymbol{\lambda}_b)\boldsymbol{Q}^\top$, can, where $\boldsymbol{Q} := \boldsymbol{Q}_a \otimes \boldsymbol{Q}_b$.

**Shampoo's estimation rule as Kronecker-based covariance estimation**   According to Claim 1 (proof in Appx. A), Shampoo's estimation rule with power $p = 1/2$ in Eq. (1) can be viewed as the optimal solution of a KL minimization problem (up to a constant scalar) when one Kronecker factor is updated independently and the other is fixed as the identity, which is known as a one-sided preconditioner (An et al., 2025; Xie et al., 2025). In practice, Shampoo further approximates the required expectations using the EMA scheme in Eq. (1).

## 3.2   IMPROVING SHAMPOO'S ESTIMATION: IDEALIZED KL-SHAMPOO

Our KL perspective reveals a key **limitation**—empirically demonstrated in Fig. 5—of Shampoo's Kronecker estimation with $p = 1/2$ as a one-sided approach: it does not adequately solve the KL-minimization problem when both factors are learned jointly. Motivated by this, we design an improved estimation rule that updates the two factors simultaneously. We term this scheme as *idealized KL-Shampoo*, which is a two-sided approach.

**Claim 2. (Idealized KL-Shampoo's covariance estimation for $\boldsymbol{S}_a$ and $\boldsymbol{S}_b$)** *The optimal solution of KL minimization* $\min_{\boldsymbol{S}_a, \boldsymbol{S}_b} \mathrm{KL}\big(\mathbb{E}[\boldsymbol{gg}^\top], \boldsymbol{S}\big)$ *with a two-sided precontioner* $\boldsymbol{S} = \boldsymbol{S}_a \otimes \boldsymbol{S}_b$ *satisfies*

$$\text{Stationarity Condition: } \boldsymbol{S}_a^* = \frac{1}{d_b}\mathbb{E}[\boldsymbol{G}\big(\boldsymbol{S}_b^*\big)^{-1}\boldsymbol{G}^\top], \quad \boldsymbol{S}_b^* = \frac{1}{d_a}\mathbb{E}[\boldsymbol{G}^\top\big(\boldsymbol{S}_a^*\big)^{-1}\boldsymbol{G}]. \quad (4)$$

---

[1] Since the current QR/eigen implementation in PyTorch does not support half-precision (bfloat16), we perform QR/eigen to compute $\boldsymbol{Q}_k$ for $k \in \{a, b\}$ in single precision (float32) and then cast and store them in half precision (bfloat16). Other matrices and vectors can be computed and updated in half precision.

**Idealized KL-Shampoo's estimation** Claim 2 (proof in Appx. B) establishes a closed-form condition (see Eq. (4)) when simultaneously learning both Kronecker factors to minimize the KL problem. In machine learning, Lin et al. (2019; 2024) treated the condition as a theoretical example of a multilinear exponential-family (see Sec. 5 of Lin et al. (2019)) for Kronecker-based optimization via natural gradient descent on matrix Gaussian, while more recently, Vyas et al. (2025b) considered a similar condition motivated heuristically by gradient whitening. However, we cannot directly use this condition due to the *correlated* update between $S_a^*$ and $S_b^*$. For example, solving $S_a^*$ requires knowing $S_b^*$ in Eq. (4) or vice versa. In practice, this condition is unachievable because the expectations in Eq. (4) must be approximated. Thus, we consider an estimated $S_k$ to approximate $S_k^*$ for $k \in \{a, b\}$ and propose an exponential moving average (EMA) scheme:

$$S_a \leftarrow (1 - \beta_2)S_a + \frac{\beta_2}{d_b}GS_b^{-1}G^\top, \quad S_b \leftarrow (1 - \beta_2)S_b + \frac{\beta_2}{d_a}G^\top S_a^{-1}G. \quad (5)$$

| | Divergence | Preconditioner Structure | Estimation Scheme |
|---|---|---|---|
| KL-Shampoo | Kullback-Leibler | Kronecker factors | MLE |
| Adafactor | von Neumann | diag. Kronecker factors | matrix diag. MME |
| Matrix Adafactor 
 (Shampoo with trace scaling) | von Neumann | Kronecker factors | matrix MME |

Table 2: Important divergences, structured SPD matrices, and equivalent Gaussian estimations in matrix cases, which can be extended to tensor cases. Under the additional zero-mean assumption, minimizing KL divergence (for SPD matrices) is equivalent to maximum-likelihood estimation (MLE) (for zero-mean Gaussian distributions), whereas minimizing VN divergence amounts to (normalized) matrix second moment matching estimation (MME) (see Sec. 3.3).

**KL-Shampoo as an MLE scheme for zero-mean Gaussian whitening** Statistically, the condition in Eq. (4) corresponds to the maximum-likelihood estimation (MLE) of a zero-mean matrix Gaussian (Dutilleul, 1999) when $\mathbb{E}[gg^\top]$ is considered as a finite average $\frac{1}{N}\sum_{i=1}^N g_i g_i^\top$. This is because MLE is equivalent to minimizing the KL divergence: $\text{KL}(\frac{1}{N}\sum_{i=1}^N g_i g_i^\top, S) = -\frac{1}{N}\sum_{i=1}^N \log \mathcal{N}(g_i; 0, S) + \text{const}$, where $g_i$ is considered as a sample generated from $\mathcal{N}(0, S)$. Thus, satisfying Eq. (4) for $S = S_a \otimes S_b$ implies that gradient matrix $G$ is generated from a zero-mean matrix Gaussian, $G \sim \mathcal{MN}(0, S_a^*, S_b^*)$, with row-wise covariance $S_a^*$ and column-wise covariance $S_b^*$ obtained by maximum likelihood. Under the condition, Shampoo-style preconditioning naturally induces matrix-Gaussian (row and column) whitening: $(S_a^*)^{-1/2} G (S_b^*)^{-1/2} \sim \mathcal{MN}(0, I_a, I_b)$. This also implies that the SOAP-like projection (i.e., gradient in the eigenbasis) amounts to covariance diagonalization under the optimal eigenbasis, as we will discuss in Sec. 5. Our KL-based approach further extends this to tensor-Gaussian whitening for tensor-valued gradients—without the prohibitive cost typically associated with SVD-based methods. Notably, this kind of whitening arises from minimizing KL divergence rather than the Frobenius norm.

**EMA scheme as a stochastic proximal gradient step for the KL minimization** Our framework allows us to further justify our EMA scheme in Eq. (5) as a stochastic proximal-gradient step (see Claim 3 and a proof in Appx. C) and establish a formal connection to the theoretical example of Lin et al. (2019; 2024). Notably, our approach uses $S^{-1/2}$ for preconditioning (Eq. (1)), following Shampoo, whereas Lin et al. (2019; 2024) propose using $S^{-1}$. A straightforward implementation of our scheme is computationally expensive, since it requires expensive matrix inversions (highlighted in red in Eq. (5)) and the slow eigendecomposition for Shampoo-type preconditioning (e.g., $S^{-1/2}$). However, these issues can be alleviated—in Sec. 4 we propose an efficient implementation.

**Claim 3.** *(KL-Shampoo's moving average scheme) The moving average scheme for $S_k$ (Eq. (5)) in idealized KL-Shampoo is a stochastic proximal-gradient step with step-size $\beta_2$ to solve the KL minimization problem in Eq. (3), for $k \in \{a, b\}$. Recall that $\beta_2$ in Eq. (5) is closely related to Adam's $\beta_2$ as $\beta_2 = 1 - \beta_2^{(Adam)}$, where $\beta_2^{(Adam)}$ is Adam's $\beta_2$.*

### 3.3 Comparison with Kronecker Schemes Using Alternative Divergences

**Unifying framework via divergence-based projection** Our perspective leads to a conceptual framework that allows us to use the following divergences. Notably, we do not require the zero-mean Gaussian assumption because we can view the minimization problem as a projection problem over SPD matrices rather than Gaussian distributions. Moreover, the Gaussian assumption may not be satisfied when using other divergences.

**Frobenius norm (F-Shampoo)** Morwani et al. (2025) consider a two-sided Shampoo variant based on the Frobenius norm and derive the optimal solution via rank-1 SVD of the second moment $\mathbb{E}[gg^\top]$ (Van Loan & Pitsianis, 1993). However, this solution is often unattainable in practice and is computationally expensive for two reasons: (1) the expectation $\mathbb{E}[gg^\top]$ must be approximated; and (2) performing the SVD is costly—yielding complexity $(O(d_a^2 d_b^2))$ in general even for rank-1 SVD—which is often higher than the eigen decompositions with complexity $(O(d_k^3))$ for $k \in \{a, b\}$ that we aim to avoid. In tensor cases, this optimal result based on SVD no longer holds. Instead, we analyze the stationarity conditions (Claim 6, Appx. F) and derive a new variant, idealized F-Shampoo (Fig. 6, Appx. F), that is structurally similar to KL-Shampoo. While a straightforward implementation of F-Shampoo performs poorly in practice, the techniques (Sec. 4) we develop for KL-Shampoo can be adapted to improve its performance (Fig. 7, Appx. H).

**Von Neumann (VN) divergence (VN-Shampoo as matrix Adafactor)** Another variant, often discussed in the literature (Morwani et al., 2025; Vyas et al., 2025a; Eschenhagen et al., 2025), is Shampoo with trace scaling. Vyas et al. (2025a) established that trace-scaled Shampoo is equivalent to running Adafactor (Shazeer & Stern, 2018) in Shampoo's eigenbasis. In contrast, KL-Shampoo is not equivalent to running Adafactor in its eigenbasis. To clarify this distinction, we make the theoretical connection between Shampoo and Adafactor more explicit: trace-scaled Shampoo is exactly a matrix generalization of Adafactor obtained by minimizing the VN divergence (Tsuda et al., 2005; Dhillon & Tropp, 2008; Nock et al., 2012) and recovers Adafactor when its Kronecker factors are restricted to be diagonal, as we establish in Claim 7 (Appx. G). Our generalization of Adafactor is also applicable in tensor cases. By contrast, KL-Shampoo minimizes the KL divergence instead of the VN divergence. A straightforward implementation of Shampoo with trace scaling—referred to as idealized VN-Shampoo—performs poorly in practice. However, the techniques we develop for KL-Shampoo in Sec. 4 can be adapted (Fig. 8, Appx. G) to substantially improve its performance so that it matches Shampoo with step-size grafting and outperforms SOAP, as shown in Figs. 10 and 11 (Appx. H). When considering the additional Gaussian assumption, KL-Shampoo is derived from the maximum likelihood principle (Sec. 3.2) while VN-Shampoo can be interpreted as matrix moment matching (Table 2). This is because row-wise matrix $2^{\text{nd}}$ moment $\mathbb{E}[GG^\top] = \frac{S_a^*}{\text{Tr}(S_b^*)}$ induces VN-Shampoo's scheme for $S_a$ (Eq. (14), Appx. G): $S_a^* = \frac{\mathbb{E}[GG^\top]}{\text{Tr}(S_b^*)}$, where we make use of the Gaussian assumption: $G \sim \mathcal{MN}(0; S_a^*, S_b^*)$. Similarly, we use column-wise matrix $2^{\text{nd}}$ moment $\mathbb{E}[G^\top G]$ to obtain the scheme for $S_b$: $S_b^* = \frac{\mathbb{E}[G^\top G]}{\text{Tr}(S_a^*)}$. This is a new interpretation that existing literature does not consider.

A natural question then arises: which divergence is more suitable? Theoretically, the KL divergence is broadly applicable to arbitrary SPD matrices (Bhatia, 2007; Boumal et al., 2014) and is widely used for covariance matrices (Minh & Murino, 2017). In contrast, the Frobenius norm does not respect the SPD constraint, and the VN divergence is primarily motivated, studied, and applied for unit-trace SPD matrices (Tsuda et al., 2005; Nielsen & Chuang, 2010). Empirically, adopting the KL divergence yields larger improvements than both the Frobenius norm and the VN divergence for designing Shampoo's schemes (see Fig. 5) and in other applications (Kulis et al., 2009).

## 4 Efficient Implementation: KL-Shampoo with QR Decomposition

We develop techniques that enable KL-Shampoo to match SOAP-level per-iteration runtime and to achieve competitive performance without step-size grafting, all without relying on eigende-composition. Vyas et al. (2025a) demonstrated that the eigendecomposition used in Shampoo's implementation (Shi et al., 2023) is more computationally expensive than QR decomposition. Motivated by this result, we aim to improve KL-Shampoo's computational efficiency by replacing the

eigendecomposition with QR decomposition. However, incorporating QR decomposition into KL-Shampoo is non-trivial because the eigenvalues of the Kronecker factors are required, and QR does not directly provide them without a significant overhead. Specifically, the eigenvalues are essential for a reduction in the computational cost of KL-Shampoo in two reasons: (1) they remove the need to compute the matrix $-1/2$ power, $\boldsymbol{S}^{-1/2} = (\boldsymbol{Q}_a \mathrm{Diag}(\boldsymbol{\lambda}_a^{\odot-1/2})\boldsymbol{Q}_a^\top) \otimes (\boldsymbol{Q}_b \mathrm{Diag}(\boldsymbol{\lambda}_b^{\odot-1/2})\boldsymbol{Q}_b^\top)$, used for KL-Shampoo's preconditioning; (2) they eliminate expensive matrix inversions in its Kronecker estimation rule (Eq. (5)), such as $\boldsymbol{S}_b^{-1} = \boldsymbol{P}_b := \boldsymbol{Q}_b \mathrm{Diag}(\boldsymbol{\lambda}_b^{\odot-1})\boldsymbol{Q}_b^\top$ in the update for $\boldsymbol{S}_a$:

$$\boldsymbol{S}_a \leftarrow (1-\beta_2)\boldsymbol{S}_a + \frac{\beta_2}{d_b}\boldsymbol{G}\boldsymbol{S}_b^{-1}\boldsymbol{G}^\top = (1-\beta_2)\boldsymbol{S}_a + \frac{\beta_2}{d_b}\boldsymbol{G}\boldsymbol{P}_b\boldsymbol{G}^\top, \qquad (6)$$

where $\boldsymbol{Q}_k$ and $\boldsymbol{\lambda}_k$ are eigenbasis and eigenvalues of $\boldsymbol{S}_k$ for $k \in \{a, b\}$, respectively.

**KL-based estimation rule for the eigenvalues $\boldsymbol{\lambda}_a$ and $\boldsymbol{\lambda}_b$ using an outdated eigenbasis**   We aim to estimate the eigenvalues using an outdated eigenbasis and replace the slow eigendecomposition with a fast QR decomposition in KL-Shampoo. Eschenhagen et al. (2025) propose estimating the eigenvalues from a Frobenius-norm perspective, using an instantaneous scheme: $\boldsymbol{\lambda}_k^{(\mathrm{inst})} := \mathrm{diag}(\boldsymbol{Q}_k^\top \boldsymbol{S}_k \boldsymbol{Q}_k)$ for $k \in \{a, b\}$. However, our empirical results (Fig. 4) indicate that this approach becomes suboptimal when an outdated eigenbasis $\boldsymbol{Q}_k$ is reused to reduce the frequency and cost of QR decompositions. In contrast, our KL perspective (see Claim 4 and its proof in Appx. D) provides a principled alternative, allowing us to use an outdated eigenbasis. Building on this claim, we introduce an exponential moving average (EMA) scheme (Step 3a of Fig. 3) for eigenvalue estimation, which can be justified as a stochastic proximal-gradient step under our KL perspective, similar to Claim 3. This scheme updates the eigenvalues at *every iteration* while updating the eigenbasis less frequently through an efficient QR-based procedure, similar to SOAP. We can view this estimation as an eigenvalue correction for using an outdated eigenbasis, as will be discussed in Sec. 5. Since it naturally scales the eigenvalues by the dimensions of the Kronecker factors, step-size grafting should not be necessary for KL-Shampoo, as argued by Eschenhagen et al. (2025) and confirmed by our empirical results (Fig. 2). Furthermore, applying this scheme enables other Shampoo variants to be competitive and even outperform SOAP, as empirically demonstrated in Figs. 7, 10 and 11 of Appx. H. These results underscore the importance of our EMA eigenvalue scheme for Shampoo-based methods.

**Claim 4.** (*Covariance estimation for eigenvalues $\boldsymbol{\lambda}_a$ and $\boldsymbol{\lambda}_b$*) *The optimal solution of KL minimization* $\min_{\boldsymbol{\lambda}_a, \boldsymbol{\lambda}_b} \mathrm{KL}\big(\mathbb{E}[\boldsymbol{g}\boldsymbol{g}^\top], \boldsymbol{S}\big)$ *with preconditioner* $\boldsymbol{S} = (\boldsymbol{Q}_a \mathrm{Diag}(\boldsymbol{\lambda}_a)\boldsymbol{Q}_a^\top) \otimes (\boldsymbol{Q}_b \mathrm{Diag}(\boldsymbol{\lambda}_b)\boldsymbol{Q}_b^\top)$ *satisfies this condition:*

$$\boldsymbol{\lambda}_a^* = \frac{1}{d_b}\mathrm{diag}\big(\boldsymbol{Q}_a^\top \mathbb{E}[\boldsymbol{G}\boldsymbol{P}_b^*\boldsymbol{G}^\top]\boldsymbol{Q}_a\big), \quad \boldsymbol{\lambda}_b^* = \frac{1}{d_a}\mathrm{diag}\big(\boldsymbol{Q}_b^\top \mathbb{E}[\boldsymbol{G}^\top \boldsymbol{P}_a^*\boldsymbol{G}]\boldsymbol{Q}_b\big), \qquad (7)$$

*where $\boldsymbol{P}_k^* := \boldsymbol{Q}_k \mathrm{Diag}\big((\boldsymbol{\lambda}_k^*)^{\odot-1}\big)\boldsymbol{Q}_k^\top$ is also defined in Eq. (6) and considered as an approximation of $\boldsymbol{S}_k^{-1}$ for $k \in \{a, b\}$ when using an outdated eigenbasis $\boldsymbol{Q} = \boldsymbol{Q}_a \otimes \boldsymbol{Q}_b$ precomputed by QR.*

## 5   INTERPRETING AND IMPROVING SOAP VIA KL MINIMIZATION

We extend the KL perspective to better understand and improve the estimation scheme used in SOAP.

**Interpreting SOAP's estimation as covariance estimation**   Recall that SOAP (Eq. (2)) applies Shampoo's scheme to estimate its Kronecker factors and then performs RMSProp (Adam) updates in the eigenbasis of these factors. Consequently, the interpretation of SOAP's Kronecker factor estimation is identical to that of Shampoo. RMSProp's second-moment estimation in the eigenbasis can itself be interpreted as the optimal solution to a separate KL divergence minimization problem, as established in Claim 5 in Appx. E. The KL perspective—distinct from the Frobenius-norm viewpoint (George et al., 2018; Eschenhagen et al., 2025)—provides a new lens for understanding RMSProp's estimation in the eigenbasis as augmented eigenvalue estimation for a covariance matrix.

**Viewing KL-Shampoo and SOAP's eigenvalue estimations as corrections for outdated eigenbasis** When an outdated eigenbasis is used, RMSProp's scheme (Step 4a of Fig. 3) for eigenvalue estimation can be viewed as a correction in an augmented (full-diagonal) eigen space, $\boldsymbol{Q}\mathrm{Diag}(\boldsymbol{d})\boldsymbol{Q}^\top$, analogous in spirit to the Frobenius-norm interpretation (Eschenhagen et al., 2025) but derived under the KL framework. This perspective also highlights a close similarity to KL-Shampoo's estimation scheme: recall that we introduced a comparable correction (Step 3a of Fig. 3) for KL-Shampoo, but in the original Kronecker-factored diagonal eigen space, $\boldsymbol{Q}\mathrm{Diag}(\boldsymbol{\lambda}_a \otimes \boldsymbol{\lambda}_b)\boldsymbol{Q}^\top$.

**(Original) Shampoo with power** $p = 1/2$ **versus Our idealized KL-Shampoo**

1: Gradient Computation $\boldsymbol{g} := \nabla\ell(\boldsymbol{\theta})$
   $\boldsymbol{G} := \mathrm{Mat}(\boldsymbol{g}) \in \mathbb{R}^{d_a \times d_b}$
2: Covariance Estimation (each iter)
$$\begin{pmatrix} \boldsymbol{S}_a \\ \boldsymbol{S}_b \end{pmatrix} \leftarrow (1 - \beta_2) \begin{pmatrix} \boldsymbol{S}_a \\ \boldsymbol{S}_b \end{pmatrix} + \beta_2 \begin{pmatrix} \Delta_a \\ \Delta_b \end{pmatrix}$$
$$\Delta_a := \begin{cases} \boldsymbol{G}\boldsymbol{G}^\top & \text{(Original)} \\ \boldsymbol{G}\boldsymbol{Q}_b \mathrm{Diag}(\boldsymbol{\lambda}_b^{\odot -1})\boldsymbol{Q}_b^\top \boldsymbol{G}^\top / d_b & \text{(KL)} \end{cases}$$
$$\Delta_b := \begin{cases} \boldsymbol{G}^\top \boldsymbol{G} & \text{(Original)} \\ \boldsymbol{G}^\top \boldsymbol{Q}_a \mathrm{Diag}(\boldsymbol{\lambda}_a^{\odot -1})\boldsymbol{Q}_a^\top \boldsymbol{G} / d_a & \text{(KL)} \end{cases}$$
3: Eigendecomposition (every $T \geq 1$ iters)
   $\boldsymbol{\lambda}_k, \boldsymbol{Q}_k \leftarrow \mathrm{eig}(\boldsymbol{S}_k)$ for $k \in \{a, b\}$
4: Preconditioning using $\boldsymbol{Q} := \boldsymbol{Q}_a \otimes \boldsymbol{Q}_b$
   $\boldsymbol{\theta} \leftarrow \boldsymbol{\theta} - \gamma(\boldsymbol{Q}\,\mathrm{Diag}(\boldsymbol{\lambda}_a \otimes \boldsymbol{\lambda}_b)^{-1/2}\boldsymbol{Q}^\top)\boldsymbol{g}$

**Replacing the slow eigendecomposition with more efficient QR updates** *(replace Step 3)*

3a: Eigenvalue Est. with EMA (**each iter**)
   (Kronecker-factored Eigenvalue $\boldsymbol{\lambda}_a \otimes \boldsymbol{\lambda}_b$)
$$\begin{pmatrix} \boldsymbol{\lambda}_a \\ \boldsymbol{\lambda}_b \end{pmatrix} \leftarrow (1 - \beta_2) \begin{pmatrix} \boldsymbol{\lambda}_a \\ \boldsymbol{\lambda}_b \end{pmatrix} + \beta_2 \begin{pmatrix} \mathrm{diag}(\boldsymbol{Q}_a^\top \Delta_a \boldsymbol{Q}_a) \\ \mathrm{diag}(\boldsymbol{Q}_b^\top \Delta_b \boldsymbol{Q}_b) \end{pmatrix}$$
3b: Infrequent Eigenbasis Estimation using QR (every $T \geq 1$ iters)
   $\boldsymbol{Q}_k \leftarrow \mathrm{qr}(\boldsymbol{S}_k \boldsymbol{Q}_k)$ for $k \in \{a, b\}$

**SOAP (ues Shampoo's eigenbasis) versus Our KL-SOAP (uses KL-Shampoo's eigenbasis): Using augmented preconditioners and QR updates** *(replace Step 4)*

4a: **Augmented** Eigenvalue Est. with EMA (**each iter**)
   (interpret RMSProp's 2nd Moment $\boldsymbol{d}$ as Augmented Eigenvalue)
   $\boldsymbol{d} \leftarrow (1 - \beta_2)\boldsymbol{d} + \beta_2 \hat{\boldsymbol{g}}^{\odot 2}$
   $\hat{\boldsymbol{g}} := \boldsymbol{Q}^\top \boldsymbol{g} = \mathrm{vec}(\boldsymbol{Q}_a^\top \boldsymbol{G} \boldsymbol{Q}_b)$
4b: Preconditioning using Augmented Eigenvalues with Eigenbasis $\boldsymbol{Q} := \boldsymbol{Q}_a \otimes \boldsymbol{Q}_b$
   $\boldsymbol{\theta} \leftarrow \boldsymbol{\theta} - \gamma(\boldsymbol{Q}\,\mathrm{Diag}(\boldsymbol{d})^{-1/2}\boldsymbol{Q}^\top)\boldsymbol{g}$
   *Equivalent to running RMSProp in Eigenbasis:*
$$\mathrm{Mat}(\boldsymbol{\theta}) \leftarrow \mathrm{Mat}(\boldsymbol{\theta}) - \gamma \boldsymbol{Q}_a \mathrm{Mat}\underbrace{\left(\frac{\hat{\boldsymbol{g}}}{\sqrt{\boldsymbol{d}}}\right)}_{\text{RMSProp}} \boldsymbol{Q}_b^\top$$

Figure 3: *Left:* Simplified Shampoo-based schemes without momentum, damping, and weight decay. See Fig. 5 for an empirical comparison and Fig. 12 for the practical KL-Shampoo. *Top Right:* For computational efficiency, we replace the eigen step with our exponential moving average (EMA) scheme to estimate eigenvalues and infrequent eigenbasis estimation using QR, where we estimate eigenvalues $\boldsymbol{\lambda}_k$ using an outdated eigenbasis $\boldsymbol{Q}_k$ for $k \in \{a, b\}$, and use the QR procedure to estimate $\boldsymbol{Q}_k$. *Bottom Right:* Simplified SOAP-based schemes without momentum. Notably, KL-SOAP needs estimation for $\boldsymbol{\lambda}_k$ in Step 3a to compute the eigenbasis $\boldsymbol{Q}$, whereas SOAP does not. Here, we view RMSProp's 2nd moment in the eigenbasis as augmented eigenvalues highlighted in blue.

**Improving SOAP's estimation**  Similar to SOAP, we propose KL-SOAP, which utilizes KL-Shampoo's estimation to update Kronecker factors and additionally employs RMSProp (Adam) in KL-Shampoo's eigenbasis. Our unified KL perspective enables us to reuse Claim 5 to justify the use of RMSProp's (Adam's) 2nd moment estimation as augmented eigenvalue estimation in KL-SOAP. Notably, when using KL-Shampoo's eigenbasis $\boldsymbol{Q}^* = \boldsymbol{Q}_a^* \otimes \boldsymbol{Q}_b^*$ obtained from the optimal condition in Eq. (4), we can see the (Gaussian) covariance of gradient $\boldsymbol{g}$ in the eigenbasis is Kronecker-diagonalized rather than fully diagonalized: $\hat{\boldsymbol{g}} = \mathrm{vec}((\boldsymbol{Q}_a^*)^\top \boldsymbol{G}\boldsymbol{Q}_b^*) = (\boldsymbol{Q}^*)^\top \boldsymbol{g} \sim \mathcal{N}(0, \mathrm{Diag}(\boldsymbol{\lambda}_a^*) \otimes \mathrm{Diag}(\boldsymbol{\lambda}_b^*))$.

## 6   EXPERIMENTAL SETUP AND EMPIRICAL EVALUATIONS

We consider four sets of experiments to demonstrate the benefits of using the KL divergence and its effectiveness. See Appx. H for the experimental setup and additional experiments.

In the first set of experiments, we demonstrate that our KL-based perspective enables a principled redesign of Shampoo, resulting in KL-Shampoo, and achieves superior performance without step-size grafting. We evaluate Shampoo with matrix powers $p = 1/2$ and $p = 1/4$, using a state-of-the-art implementation (Shi et al., 2023). As shown in Fig. 2, Shampoo requires step-size grafting to perform well, whereas KL-Shampoo performs robustly without it. Moreover, KL-Shampoo outperforms Shampoo with grafting—even in terms of step-wise progress—even when Shampoo is equipped with eigendecomposition and step-size grafting via Adam.

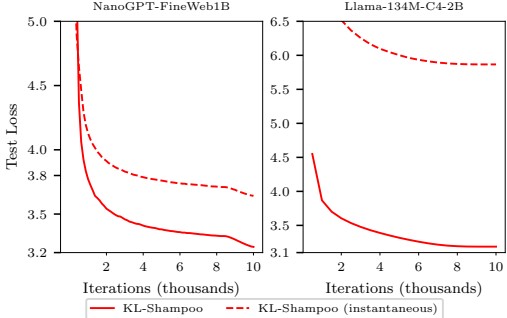 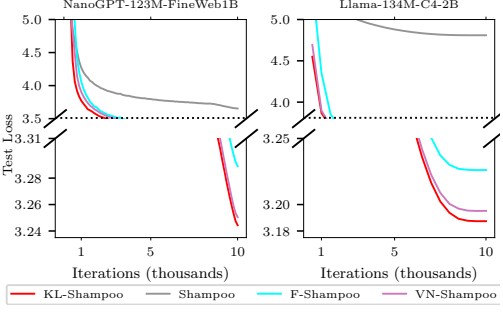

Figure 4: Empirical results (random search using 150 runs for each method) demonstrate that our EMA scheme for the eigenvalue estimation makes KL-Shampoo competitive when using an outdated eigenbasis. Without this scheme, KL-Shampoo performs poorly under an outdated eigenbasis $Q_k$ even when employing the instantaneous eigenvalue estimation $\lambda_k^{(\text{inst})} = \text{diag}(Q_k^\top S_k Q_k)$ at every iteration, as suggested by Eschenhagen et al. (2025) for $k \in \{a, b\}$. Adapting the EMA scheme also makes other variants of Shampoo competitive (Figs. 7 and 10, Appx. H) and allows the trace-scaling variant to outperform SOAP (Fig. 11, Appx. H).

Figure 5: Empirical results—based on random search with 150 runs per method—demonstrate the advantages of KL-Shampoo (two-sided) over other Shampoo variants under comparable settings, including Shampoo with $p = 1/2$ (one-sided, no grafting), F-Shampoo (two-sided, Frobenius-norm–based), and VN-Shampoo (trace scaling, two-sided von-Neumann-divergence-based). We make these variants practical by incorporating a QR step and an EMA scheme for eigenvalue estimation (Fig. 3). To ensure a fair comparison and minimize implementation bias, we implement Shampoo, F-Shampoo, and VN-Shampoo ourselves, aligning them closely with KL-Shampoo. See Fig. 11 (Appx. H) for a detailed comparison between KL-Shampoo and VN-Shampoo.

In the second set of experiments, we demonstrate that our QR-based scheme enables KL-Shampoo and KL-SOAP to achieve the same pre-iteration runtime as SOAP. We use the official SOAP implementation for comparison. As shown in Fig. 1, KL-Shampoo and KL-SOAP outperform SOAP. Remarkably, KL-Shampoo also consistently surpasses KL-SOAP while using less memory.

In the third set of experiments, we underscore the importance of using our EMA scheme for the eigenvalue estimation when working with an outdated eigenbasis. As shown in Fig. 4, the EMA scheme enables KL-Shampoo to perform well in practice, even under stale eigenbases. Moreover, this scheme can be adapted to strengthen the trace scaling variant of Shampoo (Fig. 10, Appx. H), enabling it to outperform SOAP (Fig. 11, Appx. H).

In the last set of experiments, we evaluate the benefits of using the two-sided estimation scheme under our KL perspective. Specifically, we compare the two-sided approach (KL-Shampoo) against the one-sided approach (Shampoo) in a comparable setting. To ensure fairness and eliminate implementation bias, we use our own implementation of Shampoo aligned closely with that of KL-Shampoo. For this comparison, we extend Shampoo with a QR-based step and our EMA scheme for eigenvalue estimation, as described in Fig. 3. As shown in Fig. 5, KL-Shampoo consistently and significantly outperforms Shampoo, even when Shampoo employs a similar QR-based estimation rule.

## 7 CONCLUSION

We introduced a KL perspective for interpreting Shampoo's and SOAP's structured second-moment estimation schemes. This perspective uncovers a previously unrecognized limitation of Shampoo, motivates an alternative estimation strategy to overcome it, enables a practical implementation of our approach, and extends naturally to tensor-valued estimation. Our empirical results demonstrate the effectiveness of our approach for improving the estimation schemes for Shampoo and SOAP.

## ACKNOWLEDGEMENTS

The authors would like to thank Mubarak Shah (University of Central Florida), Frank Nielsen (Sony), Mark Schmidt (University of British Columbia), and Emtiyaz Khan (RIKEN) for their helpful feedback on the manuscript, and Cheng Wu (Microsoft) and Xiya Liu (Microsoft) for their insights on language models and their support. Resources used in preparing this research were provided, in part, by the Province of Ontario, the Government of Canada through CIFAR, the companies sponsoring the Vector Institute (see `www.vectorinstitute.ai/#partners`), and Microsoft Azure ML compute.

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

## APPENDIX

## A  PROOF OF CLAIM 1

We will show that the optimal solution of KL minimization $\min_{\boldsymbol{S}_a} \mathrm{KL}\big(\mathbb{E}[\boldsymbol{g}\boldsymbol{g}^\top], \boldsymbol{S}\big)$ with a one-sided preconditioner $\boldsymbol{S} = (1/d_b\boldsymbol{S}_a) \otimes \boldsymbol{I}_b$ is $\boldsymbol{S}_a^* = \mathbb{E}[\boldsymbol{G}\boldsymbol{G}^\top]$.

By definition in Eq. (3) and substituting $\boldsymbol{S} = (1/d_b\boldsymbol{S}_a) \otimes \boldsymbol{I}_b$, we can simplify the objective function as

$$
\begin{aligned}
\mathrm{KL}&\big(\mathbb{E}[\boldsymbol{g}\boldsymbol{g}^\top], \boldsymbol{S}\big) \\
&= \frac{1}{2}\big( \log\det(\boldsymbol{S}) + \mathrm{Tr}(\boldsymbol{S}^{-1}\mathbb{E}[\boldsymbol{g}\boldsymbol{g}^\top])\big) + \text{const.} \\
&= \frac{1}{2}\big( d_b \log\det(\frac{1}{d_b}\boldsymbol{S}_a) + \mathrm{Tr}(\boldsymbol{S}^{-1}\mathbb{E}[\boldsymbol{g}\boldsymbol{g}^\top])\big) + \text{const.} \quad \text{\small (Kronecker identity for matrix det.)} \\
&= \frac{1}{2}\big( d_b \log\det(\boldsymbol{S}_a) + \mathrm{Tr}(\boldsymbol{S}^{-1}\mathbb{E}[\boldsymbol{g}\boldsymbol{g}^\top])\big) + \text{const.} \quad \text{\small (identity for a log-determinant)} \\
&= \frac{1}{2}\big( d_b \log\det(\boldsymbol{S}_a) + \mathbb{E}[\mathrm{Tr}(\boldsymbol{S}^{-1}\boldsymbol{g}\boldsymbol{g}^\top)]\big) + \text{const.} \quad \text{\small (linearity of the expectation)} \\
&= \frac{1}{2}\big( d_b \log\det(\boldsymbol{S}_a) + \mathbb{E}[\mathrm{Tr}(d_b\boldsymbol{S}_a^{-1}\boldsymbol{G}\boldsymbol{I}_b\boldsymbol{G}^\top)]\big) + \text{const.} \quad \text{\small (identity for a Kronecker vector product)} \\
&= \frac{d_b}{2}\big( \log\det(\boldsymbol{S}_a) + \mathbb{E}[\mathrm{Tr}(\boldsymbol{S}_a^{-1}\boldsymbol{G}\boldsymbol{G}^\top)]\big) + \text{const.} \\
&= \frac{d_b}{2}\big( -\log\det(\boldsymbol{P}_a) + \mathbb{E}[\mathrm{Tr}(\boldsymbol{P}_a\boldsymbol{G}\boldsymbol{G}^\top)]\big) + \text{const.}, \quad (8)
\end{aligned}
$$

where $\boldsymbol{G} = \mathrm{Mat}(\boldsymbol{g})$ and $\boldsymbol{P}_a := \boldsymbol{S}_a^{-1}$.

If we achieve the optimal solution, the stationarity condition must be satisfied regardless of the gradient with respect to $\boldsymbol{S}_a$ or $\boldsymbol{S}_a^{-1} \equiv \boldsymbol{P}_a$, such as

$$
\begin{aligned}
\boldsymbol{0} &= \partial_{\boldsymbol{S}_a^{-1}}\mathrm{KL}\big(\mathbb{E}[\boldsymbol{g}\boldsymbol{g}^\top], \boldsymbol{S}\big) \\
&= \partial_{\boldsymbol{P}_a}\mathrm{KL}\big(\mathbb{E}[\boldsymbol{g}\boldsymbol{g}^\top], \boldsymbol{S}\big) \\
&= \frac{d_b}{2}\big( -\boldsymbol{P}_a^{-1} + \mathbb{E}[\boldsymbol{G}\boldsymbol{G}^\top]\big) \quad \text{\small (use Eq. (8) and matrix calculus identities)} \\
&= \frac{d_b}{2}\big( -\boldsymbol{S}_a + \mathbb{E}[\boldsymbol{G}\boldsymbol{G}^\top]\big).
\end{aligned}
$$

Notice that the KL divergence is unbounded above. Thus, the optimal (minimal) solution exists. It must be $\boldsymbol{S}_a^* = \mathbb{E}[\boldsymbol{G}\boldsymbol{G}^\top]$ to satisfy this stationarity condition.

## B  PROOF OF CLAIM 2

We will show that the optimal solution of KL minimization $\min_{\boldsymbol{S}_a,\boldsymbol{S}_b} \mathrm{KL}\big(\mathbb{E}[\boldsymbol{g}\boldsymbol{g}^\top], \boldsymbol{S}\big)$ with a two-sided preconditioner $\boldsymbol{S} = \boldsymbol{S}_a \otimes \boldsymbol{S}_b$ should satisfy this condition: $\boldsymbol{S}_a^* = \frac{1}{d_b}\mathbb{E}[\boldsymbol{G}\big(\boldsymbol{S}_b^*\big)^{-1}\boldsymbol{G}^\top]$ and $\boldsymbol{S}_b^* = \frac{1}{d_a}\mathbb{E}[\boldsymbol{G}^\top \big(\boldsymbol{S}_a^*\big)^{-1}\boldsymbol{G}]$.

Similar to the proof of Claim 1 in Appx. A, we can simplify the objective function as

$$
\begin{aligned}
\mathrm{KL}&\big(\mathbb{E}[\boldsymbol{g}\boldsymbol{g}^\top], \boldsymbol{S}\big) \\
&= \frac{1}{2}\big( \log\det(\boldsymbol{S}) + \mathbb{E}[\mathrm{Tr}(\boldsymbol{S}^{-1}\boldsymbol{g}\boldsymbol{g}^\top)]\big) + \text{const.} \\
&= \frac{1}{2}\big( d_b \log\det(\boldsymbol{S}_a) + d_a \log\det(\boldsymbol{S}_b) + \mathbb{E}[\mathrm{Tr}(\boldsymbol{S}^{-1}\boldsymbol{g}\boldsymbol{g}^\top)]\big) + \text{const.} \text{\small (identity for a log-determinant)} \\
&= \frac{1}{2}\big( d_b \log\det(\boldsymbol{S}_a) + d_a \log\det(\boldsymbol{S}_b) + \mathbb{E}[\mathrm{Tr}(\boldsymbol{S}_a^{-1}\boldsymbol{G}\boldsymbol{S}_b^{-1}\boldsymbol{G}^\top)]\big) + \text{const.} \text{\small (identity for a Kronecker-vector-product)} \\
&= \frac{1}{2}\big( -d_b \log\det(\boldsymbol{P}_a) - d_a \log\det(\boldsymbol{P}_b) + \mathbb{E}[\mathrm{Tr}(\boldsymbol{P}_a\boldsymbol{G}\boldsymbol{P}_b\boldsymbol{G}^\top)]\big) + \text{const.}, \quad (9)
\end{aligned}
$$

where $\boldsymbol{P}_k := \boldsymbol{S}_k^{-1}$ for $k \in \{a, b\}$.

The optimal solution must satisfy the stationarity condition with respect to $\{\boldsymbol{S}_a, \boldsymbol{S}_b\}$. Notice that the gradient with respect to $\{\boldsymbol{S}_a^{-1}, \boldsymbol{S}_b^{-1}\}$ can be expressed in terms of the gradient with respect to $\{\boldsymbol{S}_a, \boldsymbol{S}_b\}$ as $\partial_{\boldsymbol{S}_a^{-1}}\mathrm{KL} = -\boldsymbol{S}_a(\partial_{\boldsymbol{S}_a}\mathrm{KL})\boldsymbol{S}_a$ and $\partial_{\boldsymbol{S}_b^{-1}}\mathrm{KL} = -\boldsymbol{S}_b(\partial_{\boldsymbol{S}_b}\mathrm{KL})\boldsymbol{S}_b$. Thus, the optimal solution must satisfy the following stationarity condition with respect to $\{\boldsymbol{S}_a^{-1}, \boldsymbol{S}_b^{-1}\}$:

$$0 = \partial_{\boldsymbol{S}_a^{-1}}\mathrm{KL}\big(\mathbb{E}[\boldsymbol{gg}^\top], \boldsymbol{S}\big), \quad 0 = \partial_{\boldsymbol{S}_b^{-1}}\mathrm{KL}\big(\mathbb{E}[\boldsymbol{gg}^\top], \boldsymbol{S}\big).$$

Using Eq. (9) and simplifying the left expression

$$\begin{aligned}
0 &= \partial_{\boldsymbol{S}_a^{-1}}\mathrm{KL}\big(\mathbb{E}[\boldsymbol{gg}^\top], \boldsymbol{S}\big) \\
&= \partial_{\boldsymbol{P}_a}\mathrm{KL}\big(\mathbb{E}[\boldsymbol{gg}^\top], \boldsymbol{S}\big) \\
&= \frac{1}{2}\big(-d_b\boldsymbol{P}_a^{-1} + \mathbb{E}[\boldsymbol{G}\boldsymbol{P}_b\boldsymbol{G}^\top]\big)
\end{aligned} \tag{10}$$

gives us this equation

$$0 = \frac{1}{2}(-d_b\boldsymbol{S}_a^* + \mathbb{E}[\boldsymbol{G}(\boldsymbol{S}_b^*)^{-1}\boldsymbol{G}^\top])$$

that the optimal solution must satisfy.

This naturally leads to the following expression:

$$\boldsymbol{S}_a^* = \frac{1}{d_b}\mathbb{E}[\boldsymbol{G}(\boldsymbol{S}_b^*)^{-1}\boldsymbol{G}^\top].$$

Likewise, we can obtain the following expression by simplifying the right expression of the stationarity condition.

$$\boldsymbol{S}_b^* = \frac{1}{d_a}\mathbb{E}[\boldsymbol{G}^\top(\boldsymbol{S}_a^*)^{-1}\boldsymbol{G}].$$

## C  PROOF OF CLAIM 3

To simplify the notation, we define $\boldsymbol{H} := \mathbf{E}[\boldsymbol{gg}^\top]$, and re-express the objective function in the KL minimization problem as $\mathcal{L}(\boldsymbol{S}) := \mathrm{KL}(\mathbf{E}[\boldsymbol{gg}^\top], \boldsymbol{S}) = \mathrm{KL}(\boldsymbol{H}, \boldsymbol{S})$. We now introduce the proximal-gradient framework (Parikh & Boyd, 2014; Khan et al., 2016) to formally state and prove Claim 3. We assume that an estimated $\boldsymbol{S}^{(t)}$ is given at iteration $t$. We use a non-negative function, $f(\boldsymbol{S}^{(t)}, \boldsymbol{S}^{(t+1)})$, to measure the closeness between the current and the next iteration. Function $f(\cdot, \cdot)$ is known as a proximal function. A (unconstrained) proximal-gradient step at iteration $t+1$ with a given proximal function, $f(\cdot, \cdot)$, is defined as the optimal solution of another minimization problem,

$$\boldsymbol{S}^{(t+1)} := \arg\min_{\boldsymbol{X}}\langle\nabla_{\boldsymbol{S}}\mathcal{L}\big|_{\boldsymbol{S}=\boldsymbol{S}^{(t)}}, \boldsymbol{X}\rangle + \frac{1}{\beta_2}f(\boldsymbol{S}^{(t)}, \boldsymbol{X}),$$

at every iteration with step-size $\beta_2$ based on the linearization of the objective function $\mathcal{L}$.

We consider a weighted quadratic function as the proximal function.

$$f(\boldsymbol{S}^{(t)}, \boldsymbol{X}) := \frac{1}{2}\|\boldsymbol{X} - \boldsymbol{S}^{(t)}\|_{\boldsymbol{W}}^2 = \frac{1}{2}\mathrm{vec}(\boldsymbol{X} - \boldsymbol{S}^{(t)})^\top\boldsymbol{W}\mathrm{vec}(\boldsymbol{X} - \boldsymbol{S}^{(t)})$$

where $\boldsymbol{W}$ is a given weight matrix. For example, $\boldsymbol{W}$ is the Hessian of the KL divergence $\boldsymbol{W} := \nabla_{\mathrm{vec}(\boldsymbol{Y})}^2\mathrm{KL}(\boldsymbol{S}^{(t)}, \boldsymbol{Y})\big|_{\boldsymbol{Y}=\boldsymbol{S}^{(t)}} = \frac{-1}{2}\big(\frac{\partial\,\mathrm{vec}(\boldsymbol{S}^{-1})}{\partial\,\mathrm{vec}(\boldsymbol{S})}\big)\big|_{\boldsymbol{S}=\boldsymbol{S}^{(t)}}$. This matrix is also known as the Fisher-Rao Riemannian metric for a zero-mean Gaussian (Amari, 2016). Note that this proximal function has been used in the quasi-Newton literature (Nocedal & Wright, 2006). Indeed, we can show that this proximal function is exactly a second-order Taylor approximation of the KL divergence, $\mathrm{KL}(\boldsymbol{S}^{(t)}, \boldsymbol{X})$, at $\boldsymbol{X} = \boldsymbol{S}^{(t)}$.

When $\boldsymbol{S} = \boldsymbol{S}_a \otimes \boldsymbol{S}_b$ admits a Kronecker product, we want to choose a weight matrix $\boldsymbol{W}$ so that this proximal function can be separated into two terms:

$$\begin{aligned}
\frac{1}{2}\|\boldsymbol{X}_a \otimes \boldsymbol{X}_b - \boldsymbol{S}^{(t)}\|_{\boldsymbol{W}}^2 &= \frac{1}{2}\|\boldsymbol{X}_a \otimes \boldsymbol{X}_b - \boldsymbol{S}_a^{(t)} \otimes \boldsymbol{S}_b^{(t)}\|_{\boldsymbol{W}}^2 \\
&= \frac{1}{2}\|\boldsymbol{X}_a - \boldsymbol{S}_a^{(t)}\|_{\boldsymbol{W}_a}^2 + \frac{1}{2}\|\boldsymbol{X}_b - \boldsymbol{S}_b^{(t)}\|_{\boldsymbol{W}_b}^2
\end{aligned}$$

Here, we consider the weight matrix as the block-diagonal Hessian of the KL divergence, such as $\boldsymbol{W} := \begin{bmatrix} \boldsymbol{W}_a & \boldsymbol{0} \\ \boldsymbol{0} & \boldsymbol{W}_b \end{bmatrix}$ by setting the cross-block terms highlighted in red to zero, where $\boldsymbol{W}_k := \partial^2_{\text{vec}(\boldsymbol{Y}_k)} \text{KL}(\boldsymbol{S}^{(t)}, \boldsymbol{Y}_a \otimes \boldsymbol{Y}_b)\big|_{\boldsymbol{Y}=\boldsymbol{S}_a^{(t)} \otimes \boldsymbol{S}_b^{(t)}}$ for $k \in \{a, b\}$. We can show that this weight matrix is exactly the block-diagonal approximation of the Fisher-Rao matrix for a zero-mean matrix Gaussian considered by Lin et al. (2019; 2024).

Now, we can formally state the claim and provide proof of it.

**Claim 3. (formal version)** The moving average scheme for $\boldsymbol{S} := \boldsymbol{S}_a \otimes \boldsymbol{S}_b$ in idealized KL-Shampoo is a proximal-gradient step at iteration $t + 1$,

$$\boldsymbol{S}_a^{(t+1)}, \boldsymbol{S}_b^{(t+1)} := \arg\min_{\boldsymbol{X}_a, \boldsymbol{X}_b} \langle \nabla_{\boldsymbol{S}_a} \mathcal{L}\big|_{\boldsymbol{S}=\boldsymbol{S}^{(t)}}, \boldsymbol{X}_a \rangle + \langle \nabla_{\boldsymbol{S}_b} \mathcal{L}\big|_{\boldsymbol{S}=\boldsymbol{S}^{(t)}}, \boldsymbol{X}_b \rangle + \frac{1}{2\beta_2} \|\boldsymbol{X}_a \otimes \boldsymbol{X}_b - \boldsymbol{S}^{(t)}\|_{\boldsymbol{W}}^2,$$

$$\iff \boldsymbol{S}_a^{(t+1)} = (1 - \beta_2)\boldsymbol{S}_a^{(t)} + \beta_2 \mathbb{E}[\boldsymbol{G}(\boldsymbol{S}_b^{(t)})^{-1}\boldsymbol{G}^\top], \quad \boldsymbol{S}_b^{(t+1)} = (1 - \beta_2)\boldsymbol{S}_b^{(t)} + \beta_2 \mathbb{E}[\boldsymbol{G}^\top(\boldsymbol{S}_a^{(t)})^{-1}\boldsymbol{G}]$$

with step-size $\beta_2$ to solve the KL minimization problem in Eq. (3), if we use a proximal function using the weight matrix, $\boldsymbol{W}$, defined above.

In mini-batch cases, we approximate the expectations using a current batch gradient (Morwani et al., 2025) (see Eq. (5)), which leads to a stochastic proximal-gradient step.

*Proof.* Because the weight matrix $\boldsymbol{W}$ is block-diagonal, we can slice this objective function for the proximal step into two terms.

$$\langle \nabla_{\boldsymbol{S}_a} \mathcal{L}\big|_{\boldsymbol{S}=\boldsymbol{S}^{(t)}}, \boldsymbol{X}_a \rangle + \langle \nabla_{\boldsymbol{S}_b} \mathcal{L}\big|_{\boldsymbol{S}=\boldsymbol{S}^{(t)}}, \boldsymbol{X}_b \rangle + \frac{1}{2\beta_2} \|\boldsymbol{X}_a \otimes \boldsymbol{X}_b - \boldsymbol{S}^{(t)}\|_{\boldsymbol{W}}^2$$

$$= \underbrace{\langle \nabla_{\boldsymbol{S}_a} \mathcal{L}\big|_{\boldsymbol{S}=\boldsymbol{S}^{(t)}}, \boldsymbol{X}_a \rangle + \frac{1}{2\beta_2} \|\boldsymbol{X}_a - \boldsymbol{S}_a^{(t)}\|_{\boldsymbol{W}_a}^2}_{(\text{block } \boldsymbol{X}_a)} + \underbrace{\langle \nabla_{\boldsymbol{S}_b} \mathcal{L}\big|_{\boldsymbol{S}=\boldsymbol{S}^{(t)}}, \boldsymbol{X}_b \rangle + \frac{1}{2\beta_2} \|\boldsymbol{X}_b - \boldsymbol{S}_b^{(t)}\|_{\boldsymbol{W}_b}^2}_{(\text{block } \boldsymbol{X}_b)}$$

Importantly, $\boldsymbol{W}_a$ and $\boldsymbol{W}_b$ are independent of $\boldsymbol{X}_a$ and $\boldsymbol{X}_b$. Thus, we solve this objective by independently for each $\boldsymbol{X}_k$ for $k \in \{a, b\}$.

We now show that solving this proximal problem gives rise to the estimation rule for $\boldsymbol{S}_a^{(t+1)}$ at iteration $t+1$. We focus on the first term since the second term does not depend on $\boldsymbol{X}_a$. We can show that $\boldsymbol{W}_a$ can be expressed as $\boldsymbol{W}_a = \partial^2_{\text{vec}(\boldsymbol{Y}_a)} \text{KL}(\boldsymbol{S}^{(t)}, \boldsymbol{Y}_a \otimes \boldsymbol{Y}_b)\big|_{\boldsymbol{Y}=\boldsymbol{S}_a^{(t)} \otimes \boldsymbol{S}_b^{(t)}} = -\frac{d_b}{2}\left(\frac{\partial \text{vec}(\boldsymbol{S}_a^{-1})}{\partial \text{vec}(\boldsymbol{S}_a)}\right)\big|_{\boldsymbol{S}=\boldsymbol{S}^{(t)}}$. This matrix $\boldsymbol{W}_a$ is also considered in Lin et al. (2024). Importantly, $\boldsymbol{W}_a$ is invertible and $\boldsymbol{W}_a^{-1} = \frac{-2}{d_b}\left(\frac{\partial \text{vec}(\boldsymbol{S}_a)}{\partial \text{vec}(\boldsymbol{S}_a^{-1})}\right)\big|_{\boldsymbol{S}=\boldsymbol{S}^{(t)}}$ With this result, the optimal solution of $\boldsymbol{X}_a$ must satisfy this stationarity condition

$$0 = \partial_{\text{vec}(\boldsymbol{X}_a)}\left(\langle \nabla_{\boldsymbol{S}_a} \mathcal{L}\big|_{\boldsymbol{S}=\boldsymbol{S}^{(t)}}, \boldsymbol{X}_a \rangle + \frac{1}{2\beta_2}\|\boldsymbol{X}_a - \boldsymbol{S}_a^{(t)}\|_{\boldsymbol{W}_a}^2\right) \text{ (note: } \|\boldsymbol{X}_a - \boldsymbol{S}_a^{(t)}\|_{\boldsymbol{W}_a}^2 := \text{vec}(\boldsymbol{X}_a - \boldsymbol{S}_a^{(t)})^\top \boldsymbol{W}_a \text{vec}(\boldsymbol{X}_a -$$

$$= \nabla_{\text{vec}(\boldsymbol{S}_a)} \mathcal{L}\big|_{\boldsymbol{S}=\boldsymbol{S}^{(t)}} + \frac{1}{\beta_2}\boldsymbol{W}_a \text{vec}(\boldsymbol{X}_a - \boldsymbol{S}_a^{(t)}) \text{ (note: } \langle \nabla_{\boldsymbol{S}_a} \mathcal{L}\big|_{\boldsymbol{S}=\boldsymbol{S}^{(t)}}, \boldsymbol{X}_a \rangle := \left(\nabla_{\text{vec}(\boldsymbol{S}_a)} \mathcal{L}\big|_{\boldsymbol{S}=\boldsymbol{S}^{(t)}}\right)^\top \text{vec}(\boldsymbol{X}_a))$$

$$\iff \text{vec}(\boldsymbol{X}_a) = \text{vec}(\boldsymbol{S}_a^{(t)}) - \beta_2 \boldsymbol{W}_a^{-1} \nabla_{\text{vec}(\boldsymbol{S}_a)} \mathcal{L}\big|_{\boldsymbol{S}=\boldsymbol{S}^{(t)}}$$

It is easy to see that the optimal solution of the proximal step is

$$
\begin{aligned}
\operatorname{vec}(\boldsymbol{S}_a^{(t+1)}) &:= \operatorname{vec}(\boldsymbol{X}_a^*) = \operatorname{vec}(\boldsymbol{S}_a^{(t)}) - \beta_2 \boldsymbol{W}_a^{-1} \nabla_{\operatorname{vec}(\boldsymbol{S}_a)} \mathcal{L}\big|_{\boldsymbol{S}=\boldsymbol{S}^{(t)}} \\
&= \operatorname{vec}(\boldsymbol{S}_a^{(t)}) - \beta_2 \underbrace{\Big(\frac{-2}{d_b}\big(\frac{\partial \operatorname{vec}(\boldsymbol{S}_a)}{\partial \operatorname{vec}(\boldsymbol{S}_a^{-1})}\big|_{\boldsymbol{S}=\boldsymbol{S}_a^{(t)}}\big)\Big)}_{=\boldsymbol{W}_a^{-1}} \nabla_{\operatorname{vec}(\boldsymbol{S}_a)} \mathcal{L}\big|_{\boldsymbol{S}=\boldsymbol{S}^{(t)}} \\
&= \operatorname{vec}(\boldsymbol{S}_a^{(t)}) + \frac{2\beta_2}{d_b} \nabla_{\operatorname{vec}(\boldsymbol{S}_a^{-1})} \mathcal{L}\big|_{\boldsymbol{S}=\boldsymbol{S}^{(t)}} \text{ \tiny (chain rule and the Jacobian matrix contained in } \boldsymbol{W}_a^{-1}\text{, which is known as Bregman duality (Lin et al., 2019))} \\
&= \operatorname{vec}(\boldsymbol{S}_a^{(t)}) + \frac{2\beta_2}{d_b} \operatorname{vec}\big( \underbrace{\big(\frac{1}{2}(-d_b \boldsymbol{S}_a^{(t)} + \mathbb{E}[\boldsymbol{G}\big(\boldsymbol{S}_b^{(t)}\big)^{-1}\boldsymbol{G}^\top])\big)}_{=\nabla_{\boldsymbol{S}_a^{-1}} \mathcal{L}|_{\boldsymbol{S}=\boldsymbol{S}^{(t)}}} \big) \text{ \tiny (recall the definition of } \mathcal{L} \text{ and use Eq. (10))} \\
&= (1-\beta_2)\operatorname{vec}(\boldsymbol{S}_a^{(t)}) + \frac{\beta_2}{d_b} \operatorname{vec}(\mathbb{E}[\boldsymbol{G}\big(\boldsymbol{S}_b^{(t)}\big)^{-1}\boldsymbol{G}^\top]),
\end{aligned}
$$

which is equivalent to the moving average scheme in Eq. (5) for updating $\boldsymbol{S}_a$ at iteration $t+1$.

Likewise, we can obtain the moving average scheme for $\boldsymbol{S}_b$. $\qquad\square$

## D   Proof of Claim 4

We will show that the optimal solution of KL minimization $\min_{\boldsymbol{\lambda}_a, \boldsymbol{\lambda}_b} \operatorname{KL}\big(\mathbb{E}[\boldsymbol{g}\boldsymbol{g}^\top], \boldsymbol{S}\big)$ with a two-sided preconditioner $\boldsymbol{S} = (\boldsymbol{Q}_a \operatorname{Diag}(\boldsymbol{\lambda}_a)\boldsymbol{Q}_a^\top) \otimes (\boldsymbol{Q}_b \operatorname{Diag}(\boldsymbol{\lambda}_b)\boldsymbol{Q}_b^\top)$ should satisfy this condition: $\boldsymbol{\lambda}_a^* = \frac{1}{d_b}\operatorname{diag}\big(\boldsymbol{Q}_a^\top \mathbb{E}[\boldsymbol{G}\boldsymbol{P}_b^*\boldsymbol{G}^\top]\boldsymbol{Q}_a\big)$ and $\boldsymbol{\lambda}_b^* = \frac{1}{d_a}\operatorname{diag}\big(\boldsymbol{Q}_b^\top \mathbb{E}[\boldsymbol{G}^\top \boldsymbol{P}_a^*\boldsymbol{G}]\boldsymbol{Q}_b\big)$, where $\boldsymbol{P}_k^* := \boldsymbol{Q}_k \operatorname{Diag}\big((\boldsymbol{\lambda}_k^*)^{\odot-1}\big) \boldsymbol{Q}_k^\top$, and $\boldsymbol{Q}_k$ is known and precomputed by QR for $k \in \{a,b\}$.

Let $\boldsymbol{S}_k := \boldsymbol{Q}_k \operatorname{Diag}(\boldsymbol{\lambda}_k)\boldsymbol{Q}_k^\top$ for $k \in \{a,b\}$. Because $\boldsymbol{Q}_k$ is orthogonal, it is easy to see that $\boldsymbol{S}_k^{-1} := \boldsymbol{Q}_k \operatorname{Diag}\big((\boldsymbol{\lambda}_k)^{\odot-1}\big)\boldsymbol{Q}_k^\top$.

Similar to the proof of Claim 2 in Appx. B, we can simplify the following objective function by substituting $\boldsymbol{S}_a$ and $\boldsymbol{S}_b$. Here, we also utilize the orthogonality of $\boldsymbol{Q}_k$ for $k \in \{a,b\}$.

$$
\begin{aligned}
&\operatorname{KL}\big(\mathbb{E}[\boldsymbol{g}\boldsymbol{g}^\top], \boldsymbol{S}\big) \\
&= \frac{1}{2}\big(d_b \log \det(\boldsymbol{S}_a) + d_a \log \det(\boldsymbol{S}_b) + \mathbb{E}[\operatorname{Tr}(\boldsymbol{S}_a^{-1}\boldsymbol{G}\boldsymbol{S}_b^{-1}\boldsymbol{G}^\top)]\big) + \text{const.} \\
&= \frac{1}{2}\big(d_b \log \det(\boldsymbol{Q}_a \operatorname{Diag}(\boldsymbol{\lambda}_a)\boldsymbol{Q}_a^\top) + d_a \log \det(\boldsymbol{Q}_b \operatorname{Diag}(\boldsymbol{\lambda}_b)\boldsymbol{Q}_b^\top) + \mathbb{E}[\operatorname{Tr}(\boldsymbol{S}_a^{-1}\boldsymbol{G}\boldsymbol{S}_b^{-1}\boldsymbol{G}^\top)]\big) + \text{const.} \\
&= \frac{1}{2}\big((d_b \sum_i \log(\lambda_a^{(i)})) + (d_a \sum_j \log(\lambda_b^{(j)})) + \mathbb{E}[\operatorname{Tr}(\boldsymbol{S}_a^{-1}\boldsymbol{G}\boldsymbol{S}_b^{-1}\boldsymbol{G}^\top)]\big) + \text{const.} \text{ \tiny (use the orthogonality of } \boldsymbol{Q}_a \text{ and } \boldsymbol{Q}_b \text{ )} \\
&= \frac{1}{2}\big((d_b \sum_i \log(\lambda_a^{(i)})) + (d_a \sum_j \log(\lambda_b^{(j)})) + \mathbb{E}[\operatorname{Tr}(\underbrace{\boldsymbol{Q}_a \operatorname{Diag}(\boldsymbol{\lambda}_a^{\odot-1})\boldsymbol{Q}_a^\top}_{=\boldsymbol{S}_a^{-1}} \boldsymbol{G} \underbrace{\boldsymbol{Q}_b \operatorname{Diag}(\boldsymbol{\lambda}_b^{\odot-1})\boldsymbol{Q}_b^\top}_{=\boldsymbol{S}_b^{-1}} \boldsymbol{G}^\top)]\big) + \text{const.}
\end{aligned}
$$
$$\tag{11}$$

The optimal $\boldsymbol{\lambda}_a$ and $\boldsymbol{\lambda}_b$ should satisfy the stationarity condition.

$$0 = \partial_{\boldsymbol{\lambda}_a}\mathrm{KL}\big(\mathbb{E}[\boldsymbol{gg}^\top], \boldsymbol{S}\big)$$

$$= \frac{1}{2}\big(d_b\boldsymbol{\lambda}_a^{\odot-1} + \partial_{\boldsymbol{\lambda}_a}\mathbb{E}[\mathrm{Tr}(\boldsymbol{Q}_a\mathrm{Diag}(\boldsymbol{\lambda}_a^{\odot-1})\boldsymbol{Q}_a^\top\boldsymbol{G}\overbrace{\boldsymbol{Q}_b\mathrm{Diag}(\boldsymbol{\lambda}_b^{\odot-1})\boldsymbol{Q}_b^\top}^{=\boldsymbol{P}_b}\boldsymbol{G}^\top)]\big) \quad \text{\small{(use Eq. (11))}}$$

$$= \frac{1}{2}\big(d_b\boldsymbol{\lambda}_a^{\odot-1} + \partial_{\boldsymbol{\lambda}_a}\mathbb{E}[\mathrm{Tr}(\mathrm{Diag}(\boldsymbol{\lambda}_a^{\odot-1})\boldsymbol{Q}_a^\top\boldsymbol{G}\boldsymbol{P}_b\boldsymbol{G}^\top\boldsymbol{Q}_a)]\big)$$

$$= \frac{1}{2}\big(d_b\boldsymbol{\lambda}_a^{\odot-1} + \partial_{\boldsymbol{\lambda}_a}\mathbb{E}[\boldsymbol{\lambda}_a^{\odot-1}\odot\mathrm{diag}\big(\boldsymbol{Q}_a^\top\boldsymbol{G}\boldsymbol{P}_b\boldsymbol{G}^\top\boldsymbol{Q}_a\big)]\big) \quad \text{\small{(utilize the trace and the diagonal structure)}}$$

$$= \frac{1}{2}\big(d_b\boldsymbol{\lambda}_a^{\odot-1} - \mathbb{E}[\boldsymbol{\lambda}_a^{\odot-2}\odot\mathrm{diag}\big(\boldsymbol{Q}_a^\top\boldsymbol{G}\boldsymbol{P}_b\boldsymbol{G}^\top\boldsymbol{Q}_a\big)]\big)$$

$$= \frac{1}{2}\big(d_b\boldsymbol{\lambda}_a^{\odot-1} - \boldsymbol{\lambda}_a^{\odot-2}\odot\mathrm{diag}\big(\boldsymbol{Q}_a^\top\mathbb{E}[\boldsymbol{G}\boldsymbol{P}_b\boldsymbol{G}^\top]\boldsymbol{Q}_a\big)\big)$$

$$\iff 0 = d_b\boldsymbol{\lambda}_a - \mathrm{diag}\big(\boldsymbol{Q}_a^\top\mathbb{E}[\boldsymbol{G}\boldsymbol{P}_b\boldsymbol{G}^\top]\boldsymbol{Q}_a\big)$$

We obtain the optimal solution by solving this equation.

$$\boldsymbol{\lambda}_a^* = \frac{1}{d_b}\mathrm{diag}\big(\boldsymbol{Q}_a^\top\mathbb{E}[\boldsymbol{G}\boldsymbol{P}_b^*\boldsymbol{G}^\top]\boldsymbol{Q}_a\big)$$

Similarly, we can obtain the other expression.

## E  PROOF OF CLAIM 5

**Claim 5.** *(SOAP and KL-SOAP's covariance estimation for augmented eigenvalues d)*  *The optimal solution of KL minimization:* $\min_{\boldsymbol{d}}\mathrm{KL}\big(\mathbb{E}[\boldsymbol{gg}^\top], \boldsymbol{S}\big)$ *with preconditioner* $\boldsymbol{S} = \boldsymbol{Q}\mathrm{Diag}(\boldsymbol{d})\boldsymbol{Q}^\top$ *is* $\boldsymbol{d}^* = \mathbb{E}\left[\big(\mathrm{vec}(\boldsymbol{Q}_a^\top\boldsymbol{G}\boldsymbol{Q}_b)\big)^{\odot 2}\right] = \mathbb{E}\left[\hat{\boldsymbol{g}}^{\odot 2}\right]$, *where* $\boldsymbol{d} \in \mathcal{R}^{d_a d_b \times 1}$ *is viewed as an augmented eigenvalue vector,* $\hat{\boldsymbol{g}} = \boldsymbol{Q}^\top\boldsymbol{g}$ *is defined at the update of (KL-)SOAP (see Eq. (2)), and* $\boldsymbol{Q} = \boldsymbol{Q}_a \otimes \boldsymbol{Q}_b$ *can be an outdated eigenbasis of (KL-)Shampoo's preconditioner.*

This proof is similar to the proof of Claim 4 in Appx. D. We will show that the optimal solution of KL minimization $\min_{\boldsymbol{d}}\mathrm{KL}\big(\mathbb{E}[\boldsymbol{gg}^\top], \boldsymbol{S}\big)$ with an augmented preconditioner $\boldsymbol{S} = (\boldsymbol{Q}\mathrm{Diag}(\boldsymbol{d})\boldsymbol{Q}^\top)$ is $\boldsymbol{d}^* = \mathbb{E}\left[\big(\mathrm{vec}(\boldsymbol{Q}_a^\top\boldsymbol{G}\boldsymbol{Q}_b)\big)^{\odot 2}\right]$, where $\boldsymbol{d} \in \mathcal{R}^{d_a d_b \times 1}$ is an augmented eigenvalue vector, $\boldsymbol{Q} := \boldsymbol{Q}_a \otimes \boldsymbol{Q}_b$, and $\boldsymbol{Q}_k$ is given and precomputed by QR for $k \in \{a, b\}$.

We can simplify the objective function by substituting $\boldsymbol{S}$. Here, we also utilize the orthogonality of $\boldsymbol{Q}_k$ for $k \in \{a, b\}$.

$$\mathrm{KL}\big(\mathbb{E}[\boldsymbol{gg}^\top], \boldsymbol{S}\big)$$

$$= \frac{1}{2}\big(\log\det(\boldsymbol{Q}\mathrm{Diag}(\boldsymbol{d})\boldsymbol{Q}^\top) + \mathrm{Tr}(\boldsymbol{Q}\mathrm{Diag}(\boldsymbol{d}^{\odot-1})\boldsymbol{Q}^\top\mathbb{E}[\boldsymbol{gg}^\top])\big) + \mathrm{const.}$$

$$= \frac{1}{2}\big(\sum_i\log(d_i)) + \mathrm{Tr}(\boldsymbol{Q}\mathrm{Diag}(\boldsymbol{d}^{\odot-1})\boldsymbol{Q}^\top\mathbb{E}[\boldsymbol{gg}^\top])\big) + \mathrm{const.} \quad \text{\small{($\boldsymbol{Q} = \boldsymbol{Q}_a \otimes \boldsymbol{Q}_b$ is orthogonal)}}$$

$$= \frac{1}{2}\big(\sum_i\log(d_i)) + \mathbb{E}\big[\mathrm{Tr}(\boldsymbol{Q}\mathrm{Diag}(\boldsymbol{d}^{\odot-1})\boldsymbol{Q}^\top\boldsymbol{gg}^\top)\big)\big] + \mathrm{const.} \quad \text{\small{(linearity of the expectation)}}$$

$$= \frac{1}{2}\big(\sum_i\log(d_i)) + \mathbb{E}\big[\mathrm{Tr}((\mathrm{vec}(\boldsymbol{Q}_a^\top\boldsymbol{G}\boldsymbol{Q}_b))^\top\mathrm{Diag}(\boldsymbol{d}^{\odot-1})\mathrm{vec}(\boldsymbol{Q}_a^\top\boldsymbol{G}\boldsymbol{Q}_b)\big] + \mathrm{const.} \text{\small{(identity of Kronecker-vector product)}}$$

$$= \frac{1}{2}\big(\sum_i\log(d_i)) + \mathbb{E}\big[\mathrm{sum}(\boldsymbol{d}^{\odot-1}\odot(\mathrm{vec}(\boldsymbol{Q}_a^\top\boldsymbol{G}\boldsymbol{Q}_b))^{\odot 2}\big] + \mathrm{const.} \text{\small{(leverage trace and diagonal struct.)}}$$

$$\tag{12}$$

The optimal $\boldsymbol{d}$ should satisfy the stationarity condition.

$$
\begin{aligned}
0 &= \partial_{\boldsymbol{d}} \mathrm{KL}\big(\mathbb{E}[\boldsymbol{g}\boldsymbol{g}^\top], \boldsymbol{S}\big) \\
&= \frac{1}{2}\big(\boldsymbol{d}^{\odot -1} - \mathbb{E}\big[\boldsymbol{d}^{\odot -2} \odot \mathrm{vec}(\boldsymbol{Q}_a^\top \boldsymbol{G}\boldsymbol{Q}_b)^{\odot 2})\big]\big) \quad \text{(use Eq. (12) and compute its derivative)} \\
\iff 0 &= \frac{1}{2}\big(\boldsymbol{d} - \mathbb{E}\big[\mathrm{vec}(\boldsymbol{Q}_a^\top \boldsymbol{G}\boldsymbol{Q}_b)^{\odot 2})\big]\big)
\end{aligned}
$$

Notice that the KL divergence is unbounded above. Thus, the optimal (minimal) solution exists and it must be $\boldsymbol{d}^* = \mathbb{E}\big[\mathrm{vec}(\boldsymbol{Q}_a^\top \boldsymbol{G}\boldsymbol{Q}_b)^{\odot 2}\big]$ to satisfy the condition.

## F TWO-SIDED SHAMPOO SCHEME BASED ON FROBENIUS NORM

**Claim 6.** *(Shampoo's estimation scheme based on Frobenius norm) The optimal solution of the Frobenius norm minimization $\min_{\boldsymbol{S}_a, \boldsymbol{S}_b} \mathrm{Frob}\big(\mathbb{E}[\boldsymbol{g}\boldsymbol{g}^\top], \boldsymbol{S}\big) := \|\mathbb{E}[\boldsymbol{g}\boldsymbol{g}^\top] - \boldsymbol{S}\|_{Frob}$ with a two-sided precontioner $\boldsymbol{S} = \boldsymbol{S}_a \otimes \boldsymbol{S}_b$ should satisfy the following condition.*

$$
\boldsymbol{S}_a^* = \frac{1}{\mathrm{Tr}((\boldsymbol{S}_b^*)^2)} \mathbb{E}[\boldsymbol{G}\boldsymbol{S}_b^* \boldsymbol{G}^\top], \quad \boldsymbol{S}_b^* = \frac{1}{\mathrm{Tr}((\boldsymbol{S}_a^*)^2)} \mathbb{E}[\boldsymbol{G}^\top \boldsymbol{S}_a^* \boldsymbol{G}], \tag{13}
$$

***Remark:*** *Although the solution can be obtained via rank-1 singular value decomposition (SVD) (Van Loan & Pitsianis, 1993) on this outer product, $\mathbb{E}[\boldsymbol{g}\boldsymbol{g}^\top]$, it can be computationally expensive to compute the solution due to the high dimensionality of the product. Moreover, the optimal solution is only achievable when the expectation of the outer product is computed exactly. Obtaining the optimal solution using SVD is even more expensive in tensor-valued cases.*

*Proof.* To simplify the proof, we will consider the square of the objective function, as the optimal solution remains unchanged. We simplify the square of the objective function by substituting $\boldsymbol{S}$. Here, we utilize the definition of the norm and re-express the norm using the matrix trace.

$$
\begin{aligned}
\|\mathbb{E}&[\boldsymbol{g}\boldsymbol{g}^\top] - \boldsymbol{S}_a \otimes \boldsymbol{S}_b\|_{\mathrm{Frob}}^2 \\
&= \mathrm{Tr}\big((\mathbb{E}[\boldsymbol{g}\boldsymbol{g}^\top] - \boldsymbol{S}_a \otimes \boldsymbol{S}_b)^\top (\mathbb{E}[\boldsymbol{g}\boldsymbol{g}^\top] - \boldsymbol{S}_a \otimes \boldsymbol{S}_b)\big) \quad \text{(an equivalent definition of the square of the norm)} \\
&= \mathrm{Tr}\big(\boldsymbol{S}_a^2 \otimes \boldsymbol{S}_b^2 - 2\mathbb{E}[\boldsymbol{g}\boldsymbol{g}^\top](\boldsymbol{S}_a \otimes \boldsymbol{S}_b)\big) + \mathrm{const.} \quad (\boldsymbol{S}_k \text{ is symmetric for } k \in \{a,b\}) \\
&= \mathrm{Tr}\big(\boldsymbol{S}_a^2\big)\mathrm{Tr}\big(\boldsymbol{S}_b^2\big) - 2\mathrm{Tr}\big(\mathbb{E}[\boldsymbol{g}\boldsymbol{g}^\top](\boldsymbol{S}_a \otimes \boldsymbol{S}_b)\big) + \mathrm{const.} \quad \text{(Property of a Kronecker product)} \\
&= \mathrm{Tr}\big(\boldsymbol{S}_a^2\big)\mathrm{Tr}\big(\boldsymbol{S}_b^2\big) - 2\mathbb{E}\big[\mathrm{Tr}\big((\boldsymbol{g}\boldsymbol{g}^\top)(\boldsymbol{S}_a \otimes \boldsymbol{S}_b)\big)\big] + \mathrm{const.} \quad \text{(linearity of the expectation)} \\
&= \mathrm{Tr}\big(\boldsymbol{S}_a^2\big)\mathrm{Tr}\big(\boldsymbol{S}_b^2\big) - 2\mathbb{E}\big[\mathrm{Tr}\big(\boldsymbol{g}^\top \mathrm{vec}(\boldsymbol{S}_a \boldsymbol{G}\boldsymbol{S}_b)\big)\big] + \mathrm{const.} \quad \text{(Property of a Kronecker product)} \\
&= \mathrm{Tr}\big(\boldsymbol{S}_a^2\big)\mathrm{Tr}\big(\boldsymbol{S}_b^2\big) - 2\mathbb{E}\big[\mathrm{Tr}\big(\boldsymbol{G}^\top \boldsymbol{S}_a \boldsymbol{G}\boldsymbol{S}_b\big)\big] + \mathrm{const.} \quad \text{(Property of a trace)}
\end{aligned}
$$

We can simplify the stationarity condition with respect to $\boldsymbol{S}_a$ as below.

$$
\begin{aligned}
0 &= \partial_{\boldsymbol{S}_a} \|\mathbb{E}[\boldsymbol{g}\boldsymbol{g}^\top] - \boldsymbol{S}_a \otimes \boldsymbol{S}_b\|_{\mathrm{Frob}}^2 \\
&= \partial_{\boldsymbol{S}_a}\big(\mathrm{Tr}\big(\boldsymbol{S}_a^2\big)\mathrm{Tr}\big(\boldsymbol{S}_b^2\big) - 2\mathbb{E}\big[\mathrm{Tr}\big(\boldsymbol{G}^\top \boldsymbol{S}_a \boldsymbol{G}\boldsymbol{S}_b\big)\big] + \mathrm{const.}\big) \\
&= 2\big(\mathrm{Tr}\big(\boldsymbol{S}_b^2\big)\boldsymbol{S}_a - \mathbb{E}[\boldsymbol{G}\boldsymbol{S}_b\boldsymbol{G}^\top]\big)
\end{aligned}
$$

Thus, the optimal solution should satisfy this condition $\boldsymbol{S}_a^* = \frac{1}{\mathrm{Tr}\big((\boldsymbol{S}_b^*)^2\big)} \mathbb{E}[\boldsymbol{G}\boldsymbol{S}_b^*\boldsymbol{G}^\top]$. Similarly, we can obtain the other condition. Morwani et al. (2025) also consider a similar condition (see Eq. 4 of their paper). $\qquad \square$

## G KEY DISTINCTION BETWEEN SHAMPOO WITH TRACE SCALING AND KL-SHAMPOO

We will show that Shampoo's estimation with trace scaling is a generalization of Adafactor. Our interpretation of Shampoo's update is grounded in a generalization of the divergence used in Adafactor—quantum relative entropy (Tsuda et al., 2005)—a Bregman divergence (Bregman, 1967) defined on



**Idealized F-Shampoo: two-sided Shampoo based on Frobenius norm** ($p = 1/2$)

1: Gradient Computation $\boldsymbol{g} := \nabla\ell(\boldsymbol{\theta})$
$\boldsymbol{G} := \mathrm{Mat}(\boldsymbol{g}) \in \mathbb{R}^{d_a \times d_b}$

2: Covariance Estimation (each iter)
$$\begin{pmatrix} \boldsymbol{S}_a \\ \boldsymbol{S}_b \end{pmatrix} \leftarrow (1-\beta_2)\begin{pmatrix} \boldsymbol{S}_a \\ \boldsymbol{S}_b \end{pmatrix} + \beta_2 \begin{pmatrix} \Delta_a \\ \Delta_b \end{pmatrix}$$

$$\Delta_a := \begin{cases} \boldsymbol{G}\boldsymbol{S}_b\boldsymbol{G}^\top / \mathrm{Tr}(\boldsymbol{S}_b^2) & \text{(v1)} \\ \boldsymbol{G}\boldsymbol{Q}_b\mathrm{Diag}(\boldsymbol{\lambda}_b)\boldsymbol{Q}_b^\top\boldsymbol{G}^\top / \sum(\boldsymbol{\lambda}_b^2) & \text{(v2)} \end{cases}$$

$$\Delta_b := \begin{cases} \boldsymbol{G}^\top\boldsymbol{S}_a\boldsymbol{G} / \mathrm{Tr}(\boldsymbol{S}_a^2) & \text{(v1)} \\ \boldsymbol{G}^\top\boldsymbol{Q}_a\mathrm{Diag}(\boldsymbol{\lambda}_a)\boldsymbol{Q}_a^\top\boldsymbol{G} / \sum(\boldsymbol{\lambda}_a^2) & \text{(v2)} \end{cases}$$

3: Eigendecomposition (every $T \geq 1$ iters)
$\boldsymbol{\lambda}_k, \boldsymbol{Q}_k \leftarrow \mathrm{eig}(\boldsymbol{S}_k)$ for $k \in \{a, b\}$

4: Preconditioning using $\boldsymbol{Q} := \boldsymbol{Q}_a \otimes \boldsymbol{Q}_b$
$\boldsymbol{\theta} \leftarrow \boldsymbol{\theta} - \gamma(\boldsymbol{Q}\,\mathrm{Diag}(\boldsymbol{\lambda}_a \otimes \boldsymbol{\lambda}_b)^{-1/2}\boldsymbol{Q}^\top)\boldsymbol{g}$





**F-Shampoo: Replacing the slow eigen step with a more efficient QR step** *(replace Step 3)*

3a: **Frequent** Eigenvalue Estimation with EMA (each iter)
$$\begin{pmatrix} \boldsymbol{\lambda}_a \\ \boldsymbol{\lambda}_b \end{pmatrix} \leftarrow (1-\beta_2)\begin{pmatrix} \boldsymbol{\lambda}_a \\ \boldsymbol{\lambda}_b \end{pmatrix} + \beta_2 \begin{pmatrix} \mathrm{diag}(\boldsymbol{Q}_a^\top \Delta_a \boldsymbol{Q}_a) \\ \mathrm{diag}(\boldsymbol{Q}_b^\top \Delta_b \boldsymbol{Q}_b) \end{pmatrix}$$

3b: Infrequent Eigenbasis Estimation using QR (every $T \geq 1$ iters)
$\boldsymbol{Q}_k \leftarrow \mathrm{qr}(\boldsymbol{S}_k\boldsymbol{Q}_k)$ for $k \in \{a, b\}$



Figure 6: *Left:* Simplified two-sided Shampoo schemes based on the Frobenius norm without momentum. We consider two variants. Variant 1 is inspired by Claim 6, while Variant 2 is similar to KL-Shampoo's update scheme, which utilizes eigenvalues. Note that Variant 1 of the idealized F-Shampoo is known as the two-sided Shampoo in the literature (Morwani et al., 2025). *Right:* Adapting our exponential moving average (EMA) approach enables F-Shampoo to use the faster QR procedure and makes it more competitive, as empirically shown in Fig. 7.

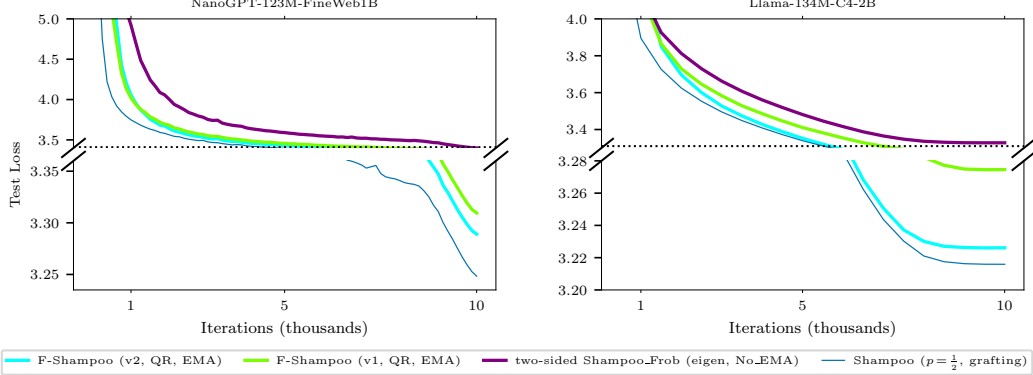

Figure 7: Empirical results from a random search with 150 runs per method on language models demonstrate that our exponential moving average (EMA) scheme for eigenvalue estimation, as described in Fig. 6, improves the performance of the two-sided Shampoo based on Frobenius norm (see Eq. 4 of Morwani et al. (2025) and Claim 6)—referred to as Variant 1 of idealized F-Shampoo. All these methods perform QR or eigen decompostion at every 10 iterations. Note that F-shampoo cannot match the performance of Shampoo with step-size grafting. This also illustrates using the Frobenius norm for preconditioner estimation is not ideal. To ensure a fair comparison and eliminate implementation bias, we use our own implementation of F-Shampoo, aligned closely with that of KL-Shampoo. As a reference, we also include the best Shampoo run with power $p = 1/2$ and grafting based on the state-of-the-art version from Meta (Shi et al., 2023).

the trace of the matrix logarithm. This new view of Shampoo's estimation is distinct from the existing Frobenius-norm perspective. By contrast, KL-Shampoo's update is based on the KL divergence (classical relative entropy)—another Bregman divergence, but one defined on the (scalar) logarithm of the matrix determinant.

We now introduce the definition of a Bregman divergence to formally discuss the distinction between Shampoo with trace scaling and KL-Shampoo. Given a strictly convex and differentiable (scalar) function $F(\cdot)$, the Bregman divergence based on this function is defined as

$$\mathcal{B}_F(\boldsymbol{X}, \boldsymbol{Y}) := F(\boldsymbol{X}) - F(\boldsymbol{Y}) - \mathrm{Tr}\big([\nabla F(\boldsymbol{Y})](\boldsymbol{X} - \boldsymbol{Y})\big).$$

As an example, the KL divergence (classical relative entropy) $\text{KL}(\boldsymbol{X}, \boldsymbol{Y})$ is a Bregman divergence with convex function $F(\boldsymbol{M}) := -\frac{1}{2}\log\det(\boldsymbol{M})$.

$$
\begin{aligned}
\mathcal{B}_F(\boldsymbol{X}, \boldsymbol{Y}) &= F(\boldsymbol{X}) - F(\boldsymbol{Y}) - \text{Tr}\big([\nabla F(\boldsymbol{Y})](\boldsymbol{X} - \boldsymbol{Y})\big) \\
&= \frac{1}{2}\big(-\log\det(\boldsymbol{X}) + \log\det(\boldsymbol{Y}) + \text{Tr}(\boldsymbol{Y}^{-1}(\boldsymbol{X} - \boldsymbol{Y}))\big) \quad \text{\small(defn. of function } F(\cdot)) \\
&= \frac{1}{2}\big(\log\det(\boldsymbol{Y}) - \log\det(\boldsymbol{X}) + \text{Tr}(\boldsymbol{Y}^{-1}\boldsymbol{X}) - \dim(\boldsymbol{X})\big) = \text{KL}(\boldsymbol{X}, \boldsymbol{Y})
\end{aligned}
$$

where $\nabla F(\boldsymbol{M}) = -\frac{1}{2}\boldsymbol{M}^{-1}$. The KL divergence is also known as the log-determinant divergence because function $F$ is defined as the logarithm of the matrix determinant. Notably, the Hessian of this $F(\cdot)$ gives rise to the Fisher-Rao metric, which is also known as the affine-invariant metric (up to a constant scalar) (Lin et al., 2023).

Now, we introduce quantum relative entropy, which is also known as von Neumann (VN) divergence, to show that Shampoo with trace scaling is a generalization of Adafactor. The VN divergence $\text{VN}(\boldsymbol{X}, \boldsymbol{Y})$ is defined as a Bregman divergence with convex function $F(\boldsymbol{M}) := \text{Tr}\big(\boldsymbol{M}\text{LogM}(\boldsymbol{M}) - \boldsymbol{M}\big)$:

$$
\begin{aligned}
\text{VN}(\boldsymbol{X}, \boldsymbol{Y}) &:= \mathcal{B}_F(\boldsymbol{X}, \boldsymbol{Y}) \\
&= F(\boldsymbol{X}) - F(\boldsymbol{Y}) - \text{Tr}\big([\nabla F(\boldsymbol{Y})](\boldsymbol{X} - \boldsymbol{Y})\big) \\
&= \text{Tr}\big(\boldsymbol{X}\text{LogM}(\boldsymbol{X}) - \boldsymbol{X} - \boldsymbol{Y}\text{LogM}(\boldsymbol{Y}) + \boldsymbol{Y} - \text{LogM}(\boldsymbol{Y})(\boldsymbol{X} - \boldsymbol{Y})\big) \quad \text{\small(defn. of function } F(\cdot)) \\
&= \text{Tr}\big(\boldsymbol{X}\text{LogM}(\boldsymbol{X}) - \boldsymbol{X} - \text{LogM}(\boldsymbol{Y})\boldsymbol{Y} + \boldsymbol{Y} - \text{LogM}(\boldsymbol{Y})(\boldsymbol{X} - \boldsymbol{Y})\big) \quad \text{\small(property of the trace)} \\
&= \text{Tr}\big(\boldsymbol{X}\text{LogM}(\boldsymbol{X}) - \boldsymbol{X} + \boldsymbol{Y} - \text{LogM}(\boldsymbol{Y})\boldsymbol{X}\big) \\
&= \text{Tr}\big(\boldsymbol{X}[\text{LogM}(\boldsymbol{X}) - \text{LogM}(\boldsymbol{Y})]\big) - \text{Tr}(\boldsymbol{X}) + \text{Tr}(\boldsymbol{Y}),
\end{aligned}
$$

where $\text{LogM}(\cdot)$ is the matrix logarithm function and Tsuda et al. (2005) show that $\nabla F(\boldsymbol{M}) = \text{LogM}(\boldsymbol{M})$. The Hessian of this $F(\cdot)$ gives rise to the Bogoliubov-Kubo-Mori (BKM) metric in quantum physics (de Boer et al., 2023).

**Claim 7.** *(**Shampoo's estimation scheme with trace scaling**) The optimal solution of the von Neumann (VN) divergence (quantum relative entropy) minimization $\min_{\boldsymbol{S}_a, \boldsymbol{S}_b} \text{VN}\big(\mathbb{E}[\boldsymbol{g}\boldsymbol{g}^\top], \boldsymbol{S}\big) := \text{Tr}(\boldsymbol{S}) - \text{Tr}\big(\mathbb{E}[\boldsymbol{g}\boldsymbol{g}^\top]\text{LogM}(\boldsymbol{S})\big) + const.$ with a two-sided precontioner $\boldsymbol{S} = \boldsymbol{S}_a \otimes \boldsymbol{S}_b$ should satisfy the following condition.*

$$
\boldsymbol{S}_a^* = \frac{1}{\text{Tr}(\boldsymbol{S}_b^*)}\mathbb{E}[\boldsymbol{G}\boldsymbol{G}^\top], \quad \boldsymbol{S}_b^* = \frac{1}{\text{Tr}(\boldsymbol{S}_a^*)}\mathbb{E}[\boldsymbol{G}^\top\boldsymbol{G}], \tag{14}
$$

*where $\text{LogM}(\cdot)$ is the matrix logarithm function.*

*The optimal solutions is Shampoo's estimation rule (power $p = \frac{1}{2}$) with trace scaling:*

$$
\boldsymbol{S}_a^* = \mathbb{E}[\boldsymbol{G}\boldsymbol{G}^\top], \quad \boldsymbol{S}_b^* = \frac{\mathbb{E}[\boldsymbol{G}^\top\boldsymbol{G}]}{\text{Tr}(\mathbb{E}[\boldsymbol{G}\boldsymbol{G}^\top])}
$$

*If we force $\boldsymbol{S}_a$ and $\boldsymbol{S}_b$ to be diagonal matrices and solve the minimization problem, we obtain Adafactor's update as shown below.*

$$
\boldsymbol{S}_a^* = \text{Diag}\big(\mathbb{E}[\boldsymbol{G}\boldsymbol{G}^\top]\big) = \text{Diag}\big(\mathbb{E}[(\boldsymbol{G}^{\odot 2})\mathbf{1}]\big)
$$

$$
\boldsymbol{S}_b^* = \text{Diag}\left(\frac{\mathbb{E}[\boldsymbol{G}^\top\boldsymbol{G}]}{\text{Tr}(\mathbb{E}[\boldsymbol{G}\boldsymbol{G}^\top])}\right) = \frac{\text{Diag}\big(\mathbb{E}[\mathbf{1}^\top\boldsymbol{G}^{\odot 2}]\big)}{\text{Tr}\big(\mathbb{E}[\mathbf{1}^\top(\boldsymbol{G}^{\odot 2})\mathbf{1}]\big)} = \frac{\text{Diag}\big(\mathbb{E}[\mathbf{1}^\top(\boldsymbol{G}^{\odot 2})]\big)}{\sqrt{\text{sum}\big(\mathbb{E}[\mathbf{1}^\top(\boldsymbol{G}^{\odot 2})]\big)\text{sum}\big(\mathbb{E}[(\boldsymbol{G}^{\odot 2})\mathbf{1}]\big)}}
$$

**Remark:** If the expectations are not computed exactly, the resulting update scheme is not the optimal solution. For example, Adafactor's update scheme is not optimal due to the EMA scheme on the diagonal Kronecker factors.

*Proof.* We will show that Shampoo's update scheme with trace scaling is an optimal solution to this minimization problem. We first simplify the objective function when $\boldsymbol{S} = \boldsymbol{S}_a \otimes \boldsymbol{S}_b$. We will use this

(Kronecker sum) identity, $\mathrm{LogM}(\boldsymbol{S}_a \otimes \boldsymbol{S}_b) = \mathrm{LogM}(\boldsymbol{S}_a) \otimes \boldsymbol{I}_b + \boldsymbol{I}_a \otimes \mathrm{LogM}(\boldsymbol{S}_b)$, to simplify the matrix logarithm.

$$
\begin{aligned}
\mathrm{VN}(\mathbb{E}[\boldsymbol{gg}^\top], \boldsymbol{S}) &= \mathrm{Tr}(\boldsymbol{S}) - \mathrm{Tr}\big(\mathbb{E}[\boldsymbol{gg}^\top]\mathrm{LogM}(\boldsymbol{S})\big) + \mathrm{const.} \\
&= \mathrm{Tr}(\boldsymbol{S}_a)\mathrm{Tr}(\boldsymbol{S}_b) - \mathrm{Tr}\big(\mathbb{E}[\boldsymbol{gg}^\top]\mathrm{LogM}(\boldsymbol{S})\big) + \mathrm{const.} \\
&= \mathrm{Tr}(\boldsymbol{S}_a)\mathrm{Tr}(\boldsymbol{S}_b) - \mathrm{Tr}\big(\mathbb{E}[\boldsymbol{gg}^\top]\big(\mathrm{LogM}(\boldsymbol{S}_a) \otimes \boldsymbol{I}_b + \boldsymbol{I}_a \otimes \mathrm{LogM}(\boldsymbol{S}_b)\big)\big) + \mathrm{const.} \\
&= \mathrm{Tr}(\boldsymbol{S}_a)\mathrm{Tr}(\boldsymbol{S}_b) - \mathrm{Tr}\big(\mathbb{E}[\boldsymbol{gg}^\top]\big(\mathrm{LogM}(\boldsymbol{S}_a) \otimes \boldsymbol{I}_b\big) + \mathrm{Tr}\big(\mathbb{E}[\boldsymbol{gg}^\top]\big(\boldsymbol{I}_a \otimes \mathrm{LogM}(\boldsymbol{S}_b)\big)\big) + \mathrm{const.} \\
&= \mathrm{Tr}(\boldsymbol{S}_a)\mathrm{Tr}(\boldsymbol{S}_b) - \mathbb{E}\big[\mathrm{Tr}\big(\boldsymbol{gg}^\top\big(\mathrm{LogM}(\boldsymbol{S}_a) \otimes \boldsymbol{I}_b\big)\big] - \mathbb{E}\big[\mathrm{Tr}\big(\boldsymbol{gg}^\top\big(\boldsymbol{I}_a \otimes \mathrm{LogM}(\boldsymbol{S}_b)\big)\big)\big] + \mathrm{const.} \\
&= \mathrm{Tr}(\boldsymbol{S}_a)\mathrm{Tr}(\boldsymbol{S}_b) - \mathbb{E}\big[\mathrm{Tr}\big(\boldsymbol{G}^\top\mathrm{LogM}(\boldsymbol{S}_a)\boldsymbol{G}\boldsymbol{I}_b\big)\big] - \mathbb{E}\big[\mathrm{Tr}\big(\boldsymbol{G}^\top\boldsymbol{I}_a\boldsymbol{G}\mathrm{LogM}(\boldsymbol{S}_b)\big)\big] + \mathrm{const.} \\
&= \mathrm{Tr}(\boldsymbol{S}_a)\mathrm{Tr}(\boldsymbol{S}_b) - \mathbb{E}\big[\mathrm{Tr}\big(\boldsymbol{G}^\top\mathrm{LogM}(\boldsymbol{S}_a)\boldsymbol{G}\big)\big] - \mathbb{E}\big[\mathrm{Tr}\big(\boldsymbol{G}^\top\boldsymbol{G}\mathrm{LogM}(\boldsymbol{S}_b)\big)\big] + \mathrm{const.} \\
&= \mathrm{Tr}(\boldsymbol{S}_a)\mathrm{Tr}(\boldsymbol{S}_b) - \mathbb{E}\big[\mathrm{Tr}\big(\boldsymbol{G}\boldsymbol{G}^\top\mathrm{LogM}(\boldsymbol{S}_a)\big)\big] - \mathbb{E}\big[\mathrm{Tr}\big(\boldsymbol{G}^\top\boldsymbol{G}\mathrm{LogM}(\boldsymbol{S}_b)\big)\big] + \mathrm{const.} \\
&= \mathrm{Tr}(\mathrm{ExpM}(\boldsymbol{P}_a))\mathrm{Tr}(\mathrm{ExpM}(\boldsymbol{P}_b)) - \mathbb{E}\big[\mathrm{Tr}\big(\boldsymbol{G}\boldsymbol{G}^\top\boldsymbol{P}_a\big)\big] - \mathbb{E}\big[\mathrm{Tr}\big(\boldsymbol{G}^\top\boldsymbol{G}\boldsymbol{P}_b\big)\big] + \mathrm{const.}
\end{aligned}
$$
(15)

where $\boldsymbol{P}_k := \mathrm{LogM}(\boldsymbol{S}_k)$ for $k \in \{a, b\}$ and $\mathrm{ExpM}(\cdot)$ is the matrix exponential function.

Notice that the optimal solution should satisfy the stationarity condition. We consider the gradient with respect to $\boldsymbol{P}_k$ because this condition must be satisfied regardless of $\boldsymbol{S}_k$ and $\boldsymbol{P}_k$ for $k \in \{a, b\}$. The condition for the derivative of Eq. (15) with respect to $\boldsymbol{P}_a$ is

$$
0 = \partial_{\boldsymbol{P}_a}\mathrm{VN}(\mathbb{E}[\boldsymbol{gg}^\top], \boldsymbol{S}) = \underbrace{\mathrm{ExpM}(\boldsymbol{P}_a)}_{=\boldsymbol{S}_a}\underbrace{\mathrm{Tr}(\mathrm{ExpM}(\boldsymbol{P}_b))}_{=\mathrm{Tr}(\boldsymbol{S}_b)} - \mathbb{E}\big[\boldsymbol{G}\boldsymbol{G}^\top\big]
$$

where Tsuda et al. (2005) show that $\partial_{\boldsymbol{P}_k}\mathrm{Tr}(\mathrm{ExpM}(\boldsymbol{P}_k)) = \mathrm{ExpM}(\boldsymbol{P}_k)$.

Thus, we can see that the optimal solution must satisfy this condition

$$
\boldsymbol{S}_a^* = \frac{\mathbb{E}\big[\boldsymbol{G}\boldsymbol{G}^\top\big]}{\mathrm{Tr}(\boldsymbol{S}_b^*)}
$$

Similarly, we can obtain the second condition.

$$
\boldsymbol{S}_b^* = \frac{\mathbb{E}\big[\boldsymbol{G}\boldsymbol{G}^\top\big]}{\mathrm{Tr}(\boldsymbol{S}_a^*)}
$$

We can verify that the following solution satisfies these conditions.

$$
\boldsymbol{S}_a^* = \mathbb{E}\big[\boldsymbol{G}\boldsymbol{G}^\top\big], \quad \boldsymbol{S}_b^* = \frac{\mathbb{E}\big[\boldsymbol{G}^\top\boldsymbol{G}\big]}{\mathrm{Tr}(\mathbb{E}\big[\boldsymbol{G}\boldsymbol{G}^\top\big])}
$$

Notice that the optimal $\boldsymbol{S}_a$ and $\boldsymbol{S}_b$ are not unique. However, their Kronecker, which is $\boldsymbol{S}^* = \boldsymbol{S}_a^* \otimes \boldsymbol{S}_b^*$, is unique. Prior studies (Morwani et al., 2025; Vyas et al., 2025a; Eschenhagen et al., 2025) have shown that this solution is an optimal Kronecker approximation of the flattened gradient second moment under the Frobenius norm.

In the Adafactor case, the result can be similarly derived when considering $\boldsymbol{S}_k$ to be a diagonal matrix for $k \in \{a, b\}$.

$\square$

## H EXPERIMENTAL SETUP AND ADDITIONAL EXPERIMENTS

**Experimental Setup** In all the experiments, we consider training four language models based on existing implementations: NanoGPT (Jordan, 2024) (123 M), NanoRWKV7 (Bo, 2024) (162 M), Llama (Glentis, 2025) (134 M), and NanoMoE (Wolfe, 2025) (227 M). We consider NanoMoE, as it contains 3D weight tensors. This model provides a natural testbed for evaluating a tensor



**Idealized VN-Shampoo: Improving Shampoo ($p = 1/2$) with trace scaling**

1: Gradient Computation $\boldsymbol{g} := \nabla\ell(\boldsymbol{\theta})$
   $\boldsymbol{G} := \mathrm{Mat}(\boldsymbol{g}) \in \mathbb{R}^{d_a \times d_b}$
2: Covariance Estimation (each iter)
$$\begin{pmatrix} \boldsymbol{S}_a \\ \boldsymbol{S}_b \end{pmatrix} \leftarrow (1 - \beta_2) \begin{pmatrix} \boldsymbol{S}_a \\ \boldsymbol{S}_b \end{pmatrix} + \beta_2 \begin{pmatrix} \Delta_a \\ \Delta_b \end{pmatrix}$$

$$\Delta_a := \begin{cases} \boldsymbol{G}\boldsymbol{G}^\top & \text{(v1)} \\ \boldsymbol{G}\boldsymbol{G}^\top / \sum(\boldsymbol{\lambda}_b) & \text{(v2)} \end{cases}$$

$$\Delta_b := \begin{cases} \boldsymbol{G}^\top\boldsymbol{G} & \text{(v1)} \\ \boldsymbol{G}^\top\boldsymbol{G} / \sum(\boldsymbol{\lambda}_a) & \text{(v2)} \end{cases}$$

3: Eigendecomposition (every $T \geq 1$ iters)
   $\boldsymbol{\lambda}_k, \boldsymbol{Q}_k \leftarrow \mathrm{eig}(\boldsymbol{S}_k)$ for $k \in \{a, b\}$
4: Preconditioning using $\boldsymbol{Q} := \boldsymbol{Q}_a \otimes \boldsymbol{Q}_b$
   $\boldsymbol{\theta} \leftarrow \boldsymbol{\theta} - \gamma(\boldsymbol{Q}\,\mathrm{Diag}(\tau\boldsymbol{\lambda}_a \otimes \boldsymbol{\lambda}_b)^{-1/2}\boldsymbol{Q}^\top)\boldsymbol{g}$

$$\tau := \begin{cases} 1/\sqrt{\mathrm{Tr}(\boldsymbol{S}_a)\mathrm{Tr}(\boldsymbol{S}_b)} & \text{(v1)} \\ 1 & \text{(v2)} \end{cases}$$





**VN-Shampoo: Replacing the slow eigen step with a more efficient QR step** *(replace Step 3)*

3a: **Frequent** Eigenvalue Estimation with EMA (each iter)
$$\begin{pmatrix} \boldsymbol{\lambda}_a \\ \boldsymbol{\lambda}_b \end{pmatrix} \leftarrow (1 - \beta_2) \begin{pmatrix} \boldsymbol{\lambda}_a \\ \boldsymbol{\lambda}_b \end{pmatrix} + \beta_2 \begin{pmatrix} \mathrm{diag}(\boldsymbol{Q}_a^\top \Delta_a \boldsymbol{Q}_a) \\ \mathrm{diag}(\boldsymbol{Q}_b^\top \Delta_b \boldsymbol{Q}_b) \end{pmatrix}$$
3b: Infrequent Eigenbasis Estimation using QR (every $T \geq 1$ iters)
   $\boldsymbol{Q}_k \leftarrow \mathrm{qr}(\boldsymbol{S}_k\boldsymbol{Q}_k)$ for $k \in \{a, b\}$



Figure 8: *Left:* Simplified VN-Shampoo schemes motivated by Claim 7 to incorporate trace scaling. We consider two variants to incorporate trace scaling into the original Shampoo. Variant 1 is inspired by Adafactor's update scheme, while Variant 2 is similar to KL-Shampoo's update scheme. Note that Variant 1 of the idealized VN-Shampoo is known as Shampoo with trace scaling in the literature. *Right:* Adapting our exponential moving average (EMA) approach enables VN-Shampoo to use the faster QR procedure and makes it competitive, as empirically shown in Fig. 10 and Fig. 11.

extension of KL-Shampoo and KL-SOAP, derived directly from our KL perspective. In doing so, we demonstrate that our methods retain the same flexibility as Shampoo and SOAP in handling tensor-valued weights without reshaping them into matrices. We train NanoGPT and NanoRWKV7 using a subset of FineWeb (1 B tokens), Llama using a subset of C4 (2 B tokens), and NanoMoE using a subset of OpenWebText (2.5 B tokens). All models except NanoMoE are trained using mini-batches with a batch size of 0.5 M. We use a batch size of 0.25 M to train NanoMoE to reduce the run time. We use the default step-size schedulers from the source implementations; NanoGPT and NanoRWKV7: linear warmup + constant step-size + linear cooldown; Llama and NanoMoE: linear warmup + cosine step-size. We tune all available hyperparameters for each method—including step-size, moving average, weight decay, damping, and momentum—using random search with 150 runs. Our hyperparameter search follows a two-stage strategy, with 75 runs in each stage. In the first stage, we search over a wider range of hyperparameters. In the second stage, we refine the search space based on the results from the first stage and focus on a narrower range. In our experiments, Shampoo by default performs eigendecomposition every 10 steps, while SOAP, KL-Shampoo, and KL-SOAP perform QR decomposition every 10 steps, as suggested by Vyas et al. (2025a).

We conduct three additional sets of experiments, following the same experimental setup as described in the main text, to further evaluate our approach. Due to limited computational resources, we focus on two language models—NanoGPT (123M) and Llama (134M)—in these additional experiments.

In the first additional experiment, we evaluate the two-sided Shampoo based on Frobenius norm (Morwani et al., 2025; Eschenhagen et al., 2025)—referred to as idealized F-Shampoo—and find that it performs poorly in practice even when we improve its performance using QR and EMA on the eigenvalues, as shown in Fig. 7. This indicates using the Frobenius norm for preconditioner estimation is not ideal.

In the second additional experiment, we evaluate Shampoo with trace scaling (Morwani et al., 2025; Vyas et al., 2025a; Eschenhagen et al., 2025)—referred to as idealized VN-Shampoo—and find that it performs poorly in practice even when using eigendecomposition. By contrast, incorporating our moving-average scheme enables it to perform well and use the fast QR decomposition, as demonstrated in Fig. 10.

In the third additional experiment, we evaluate the suitability of KL versus VN divergence for refining Shampoo's estimation rule in a comparable setting, where both variants outperform SOAP while matching SOAP-level pre-iteration runtime. As shown in Fig. 11, KL-Shampoo consistently outperforms VN-Shampoo, even when VN-Shampoo is made practical and competitive using similar

techniques to those employed in KL-Shampoo. These results underscore the advantages of the KL divergence over the VN divergence.

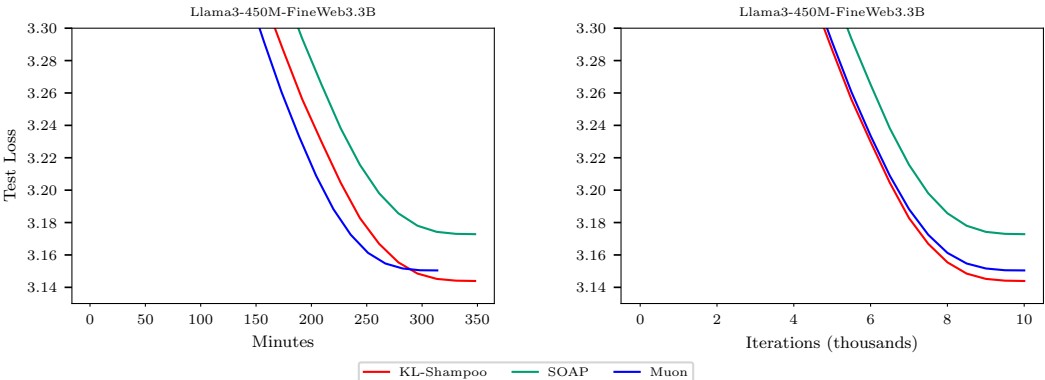

Figure 9: Empirical results (random search with 120 runs per method) show that KL-Shampoo performs better on a larger model with 450 million parameters. We perform QR every 10 iterations in this experiment using single-precision (FP32) and do not tune this frequency to optimize KL-Shampoo's runtime. From these figures, we can see that both KL-Shampoo and Muon outperform SOAP. KL-Shampoo is better than Muon (Liu et al., 2025) per iteration, but not per time. Although our results show worse performance vs. wall-clock time against Muon, we note that it is possible to improve KL-Shampoo's runtime, for example, by performing the QR decompositions less frequently and using mixed-precision QR (Higham & Mary, 2022). This is because the QR decomposition is the main computational bottleneck.

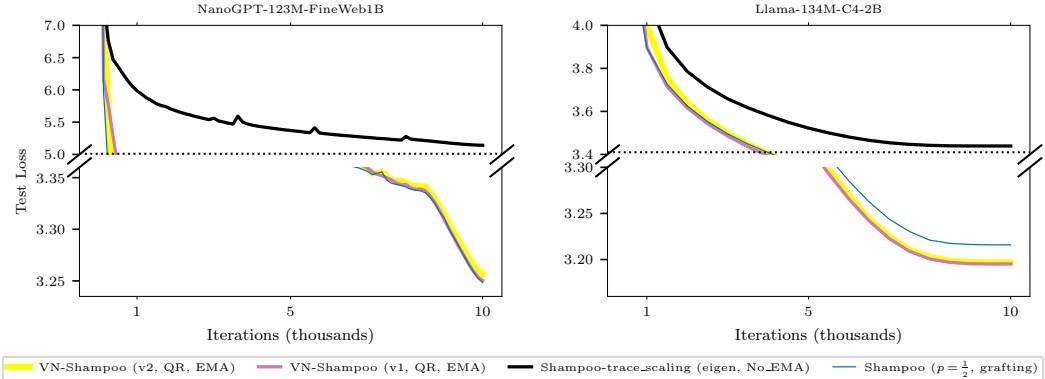

Figure 10: Empirical results from a random search with 150 runs per method on language models demonstrate that our exponential moving average (EMA) scheme for eigenvalue estimation, as described in Fig. 8, makes Shampoo with trace scaling—referred to as Variant 1 of idealized VN-Shampoo—practical and enables it to match or exceed the performance of Shampoo with step-size grafting. All these methods perform QR or eigen decompostion at every 10 iterations. Without this scheme, Shampoo with trace scaling performs poorly in practice, as shown in the figure. We implement VN-Shampoo (i.e., Shampoo with trace scaling) ourselves, as it is not available in existing implementations, including the state-of-the-art version from Meta (Shi et al., 2023). As a reference, we also include the best Shampoo run with power $p = 1/2$ and grafting based on the implementation from Meta.

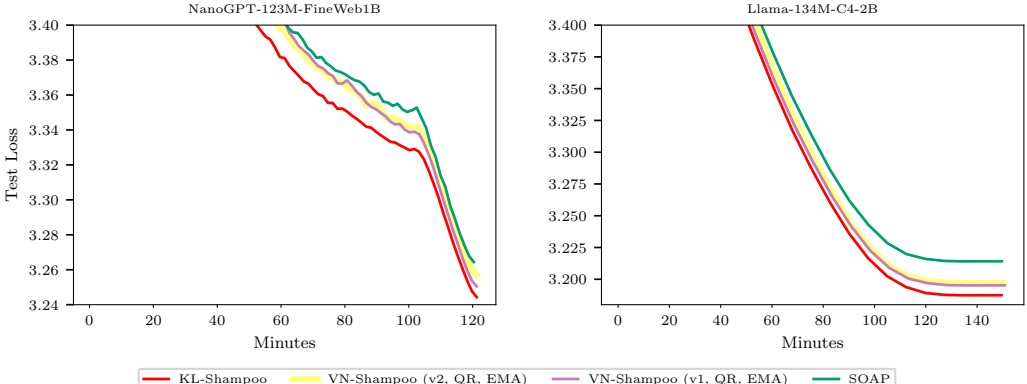

Figure 11: Empirical results (random search using 150 runs for each method) demonstrate that the advantages of KL-Shampoo over VN-Shampoo under comparable settings. In particular, we strengthen VN-Shampoo (i.e., Shampoo with trace scaling) by incorporating the QR step and the EMA scheme for eigenvalue estimation, as described in Fig. 8, to achieve SOAP-level pre-iteration runtime. To ensure a fair comparison and eliminate implementation bias, we use our own implementation of VN-Shampoo, aligned closely with that of KL-Shampoo. For runtime comparison, we include the best SOAP run as a reference. All methods take the same number of iterations in these experiments.

---

**Practical version of KL-Shampoo**

1a: Gradient Computation $\boldsymbol{g} := \nabla\ell(\boldsymbol{\theta})$
$\boldsymbol{G} := \mathrm{Mat}(\boldsymbol{g}) \in \mathbb{R}^{d_a \times d_b}$

1b: Use Gradient Momentum
$\boldsymbol{M} \leftarrow (1-\beta_1)\boldsymbol{M} + \beta_1\boldsymbol{G}$

2: Covariance Estimation (each iter)
$$\begin{pmatrix} \boldsymbol{S}_a \\ \boldsymbol{S}_b \end{pmatrix} \leftarrow (1-\beta_2)\begin{pmatrix} \boldsymbol{S}_a \\ \boldsymbol{S}_b \end{pmatrix} + \beta_2\begin{pmatrix} \Delta_a \\ \Delta_b \end{pmatrix}$$
$\Delta_a := \boldsymbol{G}\boldsymbol{Q}_b\mathrm{Diag}(\boldsymbol{\lambda}_b^{\odot-1})\boldsymbol{Q}_b^\top\boldsymbol{G}^\top/d_b = \frac{1}{d_b}[\boldsymbol{G}\boldsymbol{Q}_b\,\mathrm{Diag}(\boldsymbol{\lambda}_b^{\odot-1/2})][\boldsymbol{G}\boldsymbol{Q}_b\,\mathrm{Diag}(\boldsymbol{\lambda}_b^{\odot-1/2})]^\top$
$\Delta_b := \boldsymbol{G}^\top\boldsymbol{Q}_a\,\mathrm{Diag}(\boldsymbol{\lambda}_a^{\odot-1})\boldsymbol{Q}_a^\top\boldsymbol{G}/d_a = \frac{1}{d_a}[\boldsymbol{G}^\top\boldsymbol{Q}_a\,\mathrm{Diag}(\boldsymbol{\lambda}_a^{\odot-1/2})][\boldsymbol{G}^\top\boldsymbol{Q}_a\,\mathrm{Diag}(\boldsymbol{\lambda}_a^{\odot-1/2})]^\top$

3a: Eigenvalue Estimation with EMA (each iter)
$$\begin{pmatrix} \boldsymbol{\lambda}_a \\ \boldsymbol{\lambda}_b \end{pmatrix} \leftarrow (1-\beta_2)\begin{pmatrix} \boldsymbol{\lambda}_a \\ \boldsymbol{\lambda}_b \end{pmatrix} + \beta_2\begin{pmatrix} \mathrm{diag}(\boldsymbol{Q}_a^\top\Delta_a\boldsymbol{Q}_a) \\ \mathrm{diag}(\boldsymbol{Q}_b^\top\Delta_b\boldsymbol{Q}_b) \end{pmatrix} = (1-\beta_2)\begin{pmatrix} \boldsymbol{\lambda}_a \\ \boldsymbol{\lambda}_b \end{pmatrix} + \beta_2\begin{pmatrix} \boldsymbol{l}_a \\ \boldsymbol{l}_b \end{pmatrix}$$
$\boldsymbol{l}_a := \frac{1}{d_b}\mathrm{sum}([\boldsymbol{Q}_a^\top\boldsymbol{G}\boldsymbol{Q}_b\,\mathrm{Diag}(\boldsymbol{\lambda}_b^{\odot-1/2})]^{\odot2}, 1) = \mathrm{mean}([\boldsymbol{Q}_a^\top\boldsymbol{G}\boldsymbol{Q}_b\,\mathrm{Diag}(\boldsymbol{\lambda}_b^{\odot-1/2})]^{\odot2}, 1)$
$\boldsymbol{l}_b := \frac{1}{d_a}\mathrm{sum}([\boldsymbol{Q}_b^\top\boldsymbol{G}^\top\boldsymbol{Q}_a\,\mathrm{Diag}(\boldsymbol{\lambda}_a^{\odot-1/2})]^{\odot2}, 1) = \mathrm{mean}([\mathrm{Diag}(\boldsymbol{\lambda}_a^{\odot-1/2})\boldsymbol{Q}_a^\top\boldsymbol{G}\boldsymbol{Q}_b]^{\odot2}, 0)$

3b: Infrequent Eigenbasis Estimation using QR (every $T \geq 1$ iters)
$\boldsymbol{Q}_k \leftarrow \mathrm{qr}(\boldsymbol{S}_k\boldsymbol{Q}_k)$ for $k \in \{a, b\}$

4a: Add Weight Decay
$\boldsymbol{\theta} \leftarrow \boldsymbol{\theta} - \gamma\lambda\boldsymbol{\theta}$

4b: Preconditioning using $\boldsymbol{Q} := \boldsymbol{Q}_a \otimes \boldsymbol{Q}_b$
$\boldsymbol{\theta} \leftarrow \boldsymbol{\theta} - \gamma(\boldsymbol{Q}\,\mathrm{Diag}(\boldsymbol{\lambda}_a \otimes \boldsymbol{\lambda}_b)^{-1/2}\boldsymbol{Q}^\top)\mathrm{vec}(\boldsymbol{M})$

Figure 12: A practical version of KL-Shampoo with momentum [2] $\beta_1$ and weight decay $\lambda$. In practice, we also use either damping or clipping when computing $\boldsymbol{\lambda}_k^{\odot-1/2}$ for $k \in \{a, b\}$. In a low-precision (bfloat16) setting, we store $\boldsymbol{\lambda}_k^{\odot-1/2}$ rather than $\boldsymbol{\lambda}_k$ for $k \in \{a, b\}$ for numerical stability. In the original Shampoo, $\boldsymbol{S}_k$ is initialized by a non-zero matrix to keep eigenvalues $\boldsymbol{\lambda}_k$ non-zero. In KL-Shampoo, we directly initialize $\boldsymbol{\lambda}_k$ to be non-zero (e.g., 0.1) while keeping $\boldsymbol{S}_k$ to be zero for $k \in \{a, b\}$.

---

[2] KL-Shampoo uses momentum in the original space while KL-SOAP, like SOAP, uses momentum in the rotated space.

