# OpenReview forum: "Understanding and improving Shampoo and SOAP via Kullback-Leibler Minimization"
_ICLR.cc/2026/Conference — ICLR 2026 Poster_

### Official Review · Reviewer_Mv1d · 2025-10-25

**Soundness:** 3
**Presentation:** 3
**Contribution:** 3
**Rating:** 6
**Confidence:** 4

**Summary:**

This paper proposes KL-Shampoo and KL-SOAP, which modify the Shampoo and Soap algorithms to try to find the best approximation to the gradient second moment in the KL norm rather than the Frobenius norm. In practice this yields an update rule where current estimates of the second moment of the RHS covariance are used to update the LHS covariance and visa-versa. The paper also proposes to use QR updates, but combined with frequent estimates of eigenvalues via EMA. Empirical results show gains over the baselines.

**Strengths:**

1. The paper provides a new perspective on pre-conditioning for LLM optimization that yields principled algorithms.
2. The presentation of related work and how the KL methods fit in is very nice.
3. The method seems to work better than baselines in the experiments. And hyper parameter tuning seems relatively fair across baselines and runtime/memory is considered.

**Weaknesses:**

1. It’s a lot to cover, but the presentation of the algorithm itself could be improved. It is too nonlinear with options here and there. Some clear, linear algorithm boxes for each separate algo in the appendix could provide clarity.
2. I understand there are computational constraints, but the experiments are small-scale, so it is not clear whether the gains with scale. Perhaps even more than scaling up model size (which can get expensive), I would be interested to see substantially longer runs at the same size. The current experiments are effectively *very* early in training (less than chinchilla and far below the >>100 token-to-param rations that are more standard now), so it is not clear whether the gains matter later on as well.

**Questions:**

1. In figure 3, why do the delta definitions have two lines each?
2. In figure 3 bottom right, where is the modification that defines “augmented”?
3. Are the output layers and/or layernorms treated differently in the empirical implementation (as is often the case with similar methods)?

---

> ### Author Response · Authors · 2025-11-20
>
> We thank the reviewer for taking time to comment on our work.
> We now address the points raised by the reviewer.
>
> > the experiments are small-scale, so it is not clear whether the gains with scale.
>
> With regards to scaling up the model size, please refer to our global response about pretraining Llama3 with 450M parameters using 3.3B tokens.  This new experiment (450M) is bigger than the currently biggest experiment (227M) in the submission.
>
> With regards to scaling up the training duration, we cannot use the same tuning policy to select hyperparameters for each method due to limited compute. Our tuning policy requires at least $3 \times 4 \times 150=1800$ GPU hours per method in our current experiments ($10^4$ training steps) when using 150 runs and 4 GPUs per run for hyperparameter tuning.
> Instead, we select hyperparameters from the best run among our existing runs (2B tokens) in a new experiment. Each method takes $10^5$ training steps to train the Llama (134M) model using 20B tokens from the C4 dataset in this new experiment. Note that we can only evluate KL-Shampoo and SOAP on this model during the rebuttal phase due to limited compute.
> TL;DR: KL-Shampoo consistently outperforms SOAP when using more training steps/tokens. (see Figure 12, Appendix H).
>
> Finally, we would like to note that the focus of this paper is mostly on understanding and improving Shampoo and SOAP from a theoretical and conceptual perspective. The experiments are intended to support this through empirical observations and are not the main focus of the paper.
>
> ---
>
> > The presentation of the algorithm itself could be improved. It is too nonlinear with options here and there
>
> We agree that the presentation of the algorithm could be improved and thank the reviewer for the suggestion.
> In the algorithm box (Figure 3), we focus on highglighting connections and differences between Shampoo and KL-Shampoo, and between SOAP and KL-SOAP.
> We have modified the presentation and caption to improve the clarify of the figure (see also the point below).
> Additionally, to avoid any ambiguity in the algorithm for KL-Shampoo we have added a full version (Figure 13, Appendix H) showing its implementation, as was kindly suggested by the reviewer.
>
> > In figure 3, why do the delta definitions have two lines each?
>
> The only difference between Shampoo and KL-Shampoo is in the value of delta.
> This is highlighted with the two cases in the algorithm box, with one case (top) being Shampoo and the other (bottom) KL-Shampoo.
> We had highlighted these differences using color, but obviously this was insufficient.
> We have now updated Figure 3 to clearly distinguish the meaning of the two cases.
> We ask the reviewer to please take a look at Figure 3 in the latest version of the paper and let us know if this is now suitable.
> If not, let us know if you have any suggestions of how to further improve this algorithm box.
>
> > In figure 3 bottom right, where is the modification that defines “augmented”?
>
> What we mean by ***augmented*** eigenvalues is $d \in \mathbb{R}^{d_a d_b \times 1}$, whereas the ***regular*** eigenvalues are $\lambda_a \in \mathbb{R}^{d_a \times 1}$ and $\lambda_b \in \mathbb{R}^{d_b \times 1}$. This is because Shampoo uses $Q \mathrm{Diag}(\lambda_a \otimes \lambda_b) Q^\top$ as its preconditioner while SOAP uses $Q \mathrm{Diag}(d) Q^\top$.
> This terminology is introduced in Sec 5 and mentioned in the caption of Table 1.
> We will make sure to introduce this terminology more clearly in the paper.
>
> > Are the output layers and/or layernorms treated differently in the empirical implementation (as is often the case with similar methods)?
>
> 1d (normalization layers) and embedding parameters are trained with AdamW using the same AdamW's hyperparameters for each method (Shampoo/SOAP/Muon/KL-Shampoo/KL-SOAP).

---

### Official Review · Reviewer_TCse · 2025-10-29

**Soundness:** 4
**Presentation:** 3
**Contribution:** 3
**Rating:** 6
**Confidence:** 3

**Summary:**

The authors recast diagonal estimation as covariance estimation via KL minimization: they treat the true second moment $E[gg^\top]$ as a Gaussian covariance and seek the vector $d$ that minimizes the KL divergence between $\mathcal{N}(0, E[gg^\top])$ and $\mathcal{N}(0, Q,\mathrm{Diag}(d),Q^\top)$. They prove that the KL-optimal $d$ equals the expected squared gradient in the eigenbasis, which provides a clean justification for SOAP’s “RMSProp in an eigenbasis” update as the solution to a KL problem. From this view, the classic Kronecker estimators in Shampoo and SOAP are suboptimal for the KL objective, which motivates revised estimators. They propose KL-SOAP, which keeps SOAP’s basic pattern but replaces Shampoo’s factor estimation with KL-Shampoo’s estimation to obtain a better eigenbasis $Q$.

**Strengths:**

The paper tackles a well motivated problem and offers a clear explanation. The literature typically justifies SOAP with Frobenius norm approximations [1]. The authors replace that with a covariance estimation view that minimizes KL divergence between zero mean Gaussians. Under this view, the usual Kronecker estimators in Shampoo and SOAP do not solve the right objective when both factors are learned jointly, which motivates a two sided estimator. This reveals a limitation in the standard Kronecker update and leads to principled fixes, presented as KL Shampoo and KL SOAP. The result is better performance at roughly SOAP level iteration cost.


[1] Morwani, Depen, et al. "A New Perspective on Shampoo's Preconditioner." arXiv preprint arXiv:2406.17748 (2024).

**Weaknesses:**

The paper argues that the Frobenius-norm view misses the right geometry and proposes a KL objective instead. Since methods like SOAP ultimately aim to approximate a preconditioner, it is natural to assess estimation quality by a distance such as the Frobenius norm, and it should not be surprising that different distance functions produce different optimal approximations. I appreciate the point that reconstruction error is not the final goal, yet the justification still feels incomplete. The paper does not fully persuade me that the KL objective, rather than Frobenius or another principled choice, is the uniquely appropriate metric for evaluating structured preconditioners in this setting.



The paper presents KL-SOAP as a principled improvement to SOAP, but the mechanical change is small. In practice the algorithmic delta is limited to swapping SOAP’s Shampoo-based eigenbasis for a KL-Shampoo eigenbasis and adding an EMA rule for factor eigenvalues under basis staleness. The step form, the augmented diagonal, and the memory profile remain essentially the same. The narrative should reflect that KL-SOAP is best understood as SOAP with a stronger basis construction and more careful eigenvalue tracking, not as a distinct optimizer.

The experiments never report approximation error under the proposed KL objective.

**Questions:**

Please address my concerns above

---

> ### Author Response · Authors · 2025-11-20
>
> We thank the reviewer for taking time to comment on our work.
> We now address the points raised by the reviewer.
>
> > The paper does not fully persuade me that the KL objective, rather than Frobenius or another principled choice, is the uniquely appropriate metric for evaluating structured preconditioners in this setting.
>
> We would like to clarify that we don't claim the KL divergence to be the (uniquely) best metric in this setting. Our work provides a conceptual framework for estimating structured preconditioners, allowing us to treat different divergences in a unified way.
> For example, we can replace the KL divergence with other divergences for SPD matrices.
> We think this is a compelling advantage to our proposed framework, even beyond the use of the KL divergence.
> We do consider the Frobenius norm (new Appendix G), and the VN-divergence (resulting in a matrix version of Adafactor, Appendix F) because these divergences are already studied in the Shampoo literature. We find the KL-divergence empirically works the best among these divergences in our experiments, which motives the selection of KL-divergence from an empirical perspective.
> Additionally, the KL-divergence for SPD matrices/covariance is widely used and justified in many related fields such as SPD matrix optimization, quasi-Newton methods (like BFGS and DFP), and statistical estimation (see L206-215 of the paper), providing theoretical motivation for this choice of divergence. This does not of course exclude the possibility that other divergences may work better than the KL-divergence for this task.
>
> To further address your concern, we additionally include the two-sided Shampoo based on the Frobenius norm (referred to as F-Shampoo, see Eq 4 of Morwani et al. (2025)) in our experiments. We empirically demonstrate that KL-Shampoo outperforms VN-Shampoo, F-Shampoo, and original Shampoo in a similar setting, as shown in Figure 5. For completeness, we include a new section and additional experiments (Figure 9) in the appendix (new Appendix G) to formally discuss and evaluate F-Shampoo. Note that Morwani et al. (2025) do not evaluate F-Shampoo in their experiments and so our work now fills this gap.
>
> Finally, we add a historical note to further motivate the use of the KL divergence.  In a closely related line of work, the standard Frobenius norm was originally used in the 1970s to estimate a quasi-Newton-based SPD preconditioner [3,4], and was later replaced in the 1990s by the KL divergence (or by a weighted norm corresponding to the second-order truncation of the KL divergence (see Sec 6.1 of Nocedal & Wright (2006))). Consequently, the well-known and widely used quasi-Newton methods BFGS and DFP can be derived as solutions of an untruncated KL minimization problem (Byrd & Nocedal, 1989; Fletcher, 1991). In particular, using the standard Frobenius norm does not recover the BFGS and DFP preconditioner estimation schemes.
>
> > The paper presents KL-SOAP as a principled improvement to SOAP, but the mechanical change is small. [...] The narrative should reflect that KL-SOAP is best understood as SOAP with a stronger basis construction and more careful eigenvalue tracking, not as a distinct optimizer.
>
> Thank you for your suggestion. We will make sure to highlight KL-SOAP as modification of SOAP which uses a 'refined' basis estimation algorithm based on KL-Shampoo.
> Just for clarification, our contribution about KL-SOAP is that we interpret the Adam/RMSProp second moment estimation in KL-SOAP as an augmented eigenvalue estimation, and a correction when using an outdated eigenbasis.
>
>
> > The experiments never report approximation error under the proposed KL objective.
>
> We could show a plot where the KL loss goes down, but unfortunately this would only be feasible for a small model and thus may have limited utility.
> Computing the exact KL loss in a high-dimensional case is challenging. For example, given a $10^2$-by-$10^2$ weight matrix, its flattened gradident second moment $\mathbb{E}[gg^\top]$ is a $10^4$-by-$10^4$ matrix, where the flattened gradient $g \in \mathcal{R}^{10^4}$.
> Also, note that this is not our main focus of this paper. We focus on using the KL estimation scheme to improve (Shampoo's) preconditioning and verifying the scheme in experiments similar to the ones considered in the Adafactor and SOAP papers.
> Moreover, we have theoretically shown that KL-Shampoo's estimation rule is a proximal gradient descent step to solve the KL objective (see Claim 3).
>
> ---
> Reference:
>
> [3] Dennis, Jr, John E., and Jorge J. Moré. "Quasi-Newton methods, motivation and theory." SIAM review 19.1 (1977): 46-89.
>
> [4] John Greenstadt. Variations on variable-metric methods. In SIAM REVIEW, volume 10, pp. 474, 1968.

---

### Official Review · Reviewer_1qBq · 2025-10-31

**Soundness:** 4
**Presentation:** 3
**Contribution:** 4
**Rating:** 8
**Confidence:** 4

**Summary:**

The paper proposes a KL-divergence perspective of looking at the updates provided by Shampoo and SOAP. It provides a new update rule for the kronecker factored preconditioner based on the KL divergence perspective and provides empirical results for the same. Empirically, KL-Shampoo and KL-SOAP outperform Shampoo and SOAP on various language modeling benchmarks.

**Strengths:**

1. The paper provides a novel perspective of viewing the Kronecker factorization updates based on KL-divergence.

2. It provides a new update rule for the kronecker factor updates based on this perspective, and also provides a computationally efficient way for implementing it.

3. The empirical results of the method are pretty strong, outperforming recent second order methods such as Shampoo and SOAP.

**Weaknesses:**

1. The paper is missing comparisons to Muon, another popular second order optimizer proposed recently.

**Questions:**

1. Can the authors add comparison to Muon as well - for both runtime and number of iterations?

---

> ### Author Response · Authors · 2025-11-20
>
> We graciously thank the reviewer for their thoughtful feedback, and strong support of the work.
>
> ---
>
> > missing comparisons to Muon
>
> Please see our global response. We have now added a comparison with Muon, which we hope is sufficient to address this concern.

---

### Official Review · Reviewer_ENoC · 2025-10-31

**Soundness:** 3
**Presentation:** 3
**Contribution:** 3
**Rating:** 6
**Confidence:** 2

**Summary:**

The paper reinterprets the preconditioner estimation in Shampoo and SOAP as a covariance estimation problem solved via Kullback-Leibler (KL) divergence minimization.  This perspective may reveal a previously overlooked limitation in the one-sided estimation approach of Shampoo. Building on this insight, the authors propose new estimation rules, leading to KL-Shampoo and KL-SOAP. Through  experiments on multiple language models, the authors demonstrate that KL-Shampoo and KL-SOAP outperform Shampoo and SOAP.

**Strengths:**

- The KL divergence view is a well-motivated approach. The connection to covariance estimation and the justification via proximal-gradient steps are interesting.

- The proposed methods are not just theoretical constructs; they are practical algorithms that deliver comparable performance over Shampoo and SOAP.

- The paper is generally well-written.

**Weaknesses:**

- The authors only demonstrate that KL-Shampoo and KL-SOAP outperform Shampoo and SOAP. They do not compare with other state-of-the-art large scale algorithms.

- The authors do not provide any theoretical convergence guarantees.

**Questions:**

Page 3, Figure 1 Caption: "pre-iteration runtime",  "per-iteration runtime"?


Appendix G: It might be helpful to briefly summarize the conclusions of Fig. 7 and Fig. 8 in the main text, as they provide strong support for the choice of KL divergence over VN divergence.

Could the authors provide theoretical guarantees, showing the advantage of the proposed algorithms?

Could the authors provide more numerical results to compare with other emerging large scale algorithms?

---

> ### Author Response · Authors · 2025-11-20
>
> We thank the reviewer for taking time to comment on our work.
> We now address the points raised by the reviewer.
>
> > [The authors] do not compare with other state-of-the-art large scale algorithms.
>
> Please see our global response. We have now added a comparsion with Muon, which is a SOTA algorithm with recent popularity.  Due to limited compute, we cannot add more methods without properly and fairly tuning them.
>
> > Appendix G: It might be helpful to briefly summarize the conclusions of Fig. 7 and Fig. 8 in the main text, as they provide strong support for the choice of KL divergence over VN divergence.
>
> We thank the reviewer for the useful suggestion.  We have adjusted the main text by including VN-Shampoo in Figure 5 and updated the caption correspondingly, and updated Sec 6 of the main text to include some discussion on the advantages of KL-divergence over VN-divergence.
>
>
> > The authors do not provide any theoretical convergence guarantees.
>
> While we agree with the reviewer that deriving rigorous bounds for our algorithms would be useful, doing convergence analysis under realistic assumptions for LLM training itself is highly nontrivial and would be a significant undertaking. Thus, similar to the SOAP paper, we will leave this to future work.

---

### Author Response · Authors · 2025-11-20
**Global response**

> Comparison with other state-of-the-art large-scale algorithms (Reviewer ENoC)

> Missing comparisons to Muon (Reviewer 1qBq)

> The experiments are small-scale, so it is not clear whether the gains with scale. (Reviewer Mv1d)

We evaluate KL-Shampoo, SOAP, and Muon [1] on a new, larger, model (Llama3-450M) using 100 runs per method.  This allows us to reduce the computation cost while properly and fairly tuning Muon.  We can only evaluate Muon on this model (and not the other models already in the paper) using 100 runs (rather than 150 runs) during the rebuttal phase due to limited compute resources. This experiment (450M) is larger than the largest experiment (227M) currently in the submission.

The plots about this experiment can be found at Figure 11, Appendix H.
TL;DR: KL-Shampoo is better than Muon [1] per-iteration, but not per-time (10% overhead when performing QR in FP32 every 10 iterations) for pre-training Llama3-450M.

Although our results show worse performance vs wall-clock time against Muon, we note that it is possible to improve KL-Shampoo's runtime, for example by performing the QR decompositions less frequently and using mixed-precision QR [2]. This is because the QR decomposition is the main computational bottleneck.
We do not tune the decomposition frequency to optimize KL-Shampoo's runtime.
There are additional algorithmic modifications and engineering efforts that can further improve KL-Shampoo's runtime. We leave this for future work.

Finally, we would like to highlight that the focus of this paper is on understanding and improving Shampoo and SOAP—optimizers that naturally support tensor-valued weights. Thus, we did not initially include Muon, which is a specialized method for matrix-valued weights (i.e., 2D tensors), in the paper. Shampoo and SOAP are more general than Muon, and correspondingly so are KL-Shampoo and KL-SOAP.

Reference:

[1] Muon with an adjusted step-size; Liu et al, "Muon is scalable for LLM training", arXiv preprint arXiv:2502.16982 (2025).

[2] Higham, Nicholas J., and Theo Mary. "Mixed precision algorithms in numerical linear algebra." Acta Numerica 31 (2022): 347-414.

---

### Author Response · Authors · 2025-11-20
**New changes in the latest version (November 21)**

* Main text
    * improved the presentation of the algorithm box in Figure 3  (Reviewer Mv1d)
    * updated Figure 5 to put KL-Shampoo, original Shampoo (without grafting), VN-Shampoo (matrix version of Adafactor), and F-Shampoo (two-sided Shampoo based on the Frobenius norm) side by side and demonstrate the advantages of KL-Shampoo (Reviewers ENoC and TCse)
    * included a short discussion in Section 6 on the advantages of using the KL-divergence over the VN-divergence  (Reviewer ENoC)
* The appendix
    * added a new Section (Appendix G) and  additional experiments (Figure 9, Appendix G) about Shampoo based on the Frobenius norm (i.e., F-Shampoo) for completeness (Reviewer TCse)
    * included an individual algorithm box for KL-Shampoo (Figure 13, Appendix H) in the appendix (Reviewer Mv1d)
    * added a new experiment (Figure 11, Appendix H) on Llama3-450M to include Muon as a strong baseline (Reviewers ENoC, 1qBq, Mv1d)
    * added a new experiment (Figure 12, Appendix H) on Llama-134M using 20B tokens to support the advantages of KL-Shampoo when using larger training steps/tokens (Reviewer Mv1d)
    * fixed some typos in Appendix C

---

### Meta-Review · Area_Chair_KPev · 2026-01-06

**Summary:**

This paper offers a new perspective that interprets the estimation procedures in Shampoo and SOAP
as instances of covariance estimation using KL divergence minimization. The reviewers appreciate the contributions of this paper, including KL Shampoo and KL SOAP which are comparable with Shamoo and SOAP.

**Reviewer Concerns:**

Most reviewers commented that the paper did not have theoretical convergence, the experiments are small scale, and comparions with Muon are suggested. The authors added experiments to address this, while theory should be for future work.

**Reviewer Scores:**

Reviewer scores are initially good.

---

### Decision · Program_Chairs · 2026-01-26

Accept (Poster)